# Balanced LoRA: Removing Parameter Invariance to Accelerate Convergence

Valérie Castin [1 2]   Kimia Nadjahi [1 2]   Pierre Ablin [3]   Gabriel Peyré [1 2]

## Abstract

Low-Rank Adaptation (LoRA) is the most widely adopted method for fine-tuning large language models. Notably, LoRA is inherently overparameterized: multiple pairs of low-rank factors can yield the same adapted weight matrix. We show—both theoretically and empirically—that these pairs exhibit significantly different condition numbers. As a result, converging to different loss minimizers directly impacts the convergence rate of LoRA. Building on this observation, we introduce Balanced Low-Rank Adaptation (BaLoRA), a variant of LoRA that projects iterates onto a balanced manifold. This manifold improves the conditioning of the loss landscape while preserving the adapted matrix. The projection step is computationally lightweight and integrates seamlessly into existing fine-tuning pipelines. Empirically, BaLoRA converges faster than standard LoRA and achieves superior performance across a range of fine-tuning tasks.

## 1. Introduction

Pretrained foundation models are now ubiquitous in natural language processing (Brown et al., 2020; Qin et al., 2023; Taori et al., 2023b), computer vision (Awais et al., 2025), and multimodal learning (Li et al., 2022; Liu et al., 2023a), thanks to their ability to generalize from large-scale, diverse training data. Their massive number of parameters allows them to capture a wide range of patterns, making them ideal bases for building specialized fine-tuned models on task-specific data. However, as model sizes expand, full fine-tuning (updating all parameters) becomes impractical due to its prohibitive computational and memory costs.

To address this issue, parameter-efficient fine-tuning (PEFT)

[1]École Normale Supérieure PSL, Paris, France [2]CNRS [3]Apple, Paris, France. Correspondence to: Valérie Castin <valerie.castin@ens.fr>.

*Proceedings of the 43rd International Conference on Machine Learning*, Seoul, South Korea. PMLR 306, 2026. Copyright 2026 by the author(s).

methods have become increasingly popular (Houlsby et al., 2019). They leverage the observation that overparameterized models often exhibit low intrinsic dimensionality (Li et al., 2018; Aghajanyan et al., 2021). These methods adapt large pretrained models by updating only a small subset of parameters, reducing computational costs while preserving performance. Among them, Low-Rank Adaptation (LoRA) (Hu et al., 2022) stands out as one of the most effective approaches. Rather than updating dense weight matrices, LoRA uses trainable low-rank matrices added to the frozen pretrained weights. Specifically, a pretrained weight matrix $W \in \mathbb{R}^{a \times b}$ is updated as $W + AB$, where $A \in \mathbb{R}^{a \times r}$, $B \in \mathbb{R}^{r \times b}$, and $r \ll \min(a, b)$ is the LoRA rank. By freezing $W$, this approach reduces the number of trainable parameters from $a \times b$ (full fine-tuning) to $r \times (a + b)$, achieving substantial memory savings while preserving adaptability.

Given its empirical success across diverse applications, LoRA has recently sparked theoretical interest, though still in its early stages. Recent studies have explored its expressivity in feedforward neural networks and transformers (Zeng & Lee, 2024), analyzed its fine-tuning dynamics in the Neural Tangent Kernel (NTK) regime (Malladi et al., 2023; Jang et al., 2024), examined the distinct roles of the $A$ and $B$ matrices (Zhu et al., 2024; Hayou et al., 2024b), and investigated the effects of various initialization strategies (Hayou et al., 2024a; Li et al., 2025).

In this paper, we analyze the asymptotic behavior of training dynamics of LoRA. Specifically, we bound the asymptotic convergence rate of LoRA when fine-tuning one layer of a deep, possibly non-linear, neural network, by establishing tight bounds on the loss condition number at convergence. Our study hinges on a key property of LoRA: its overparameterization. More precisely, for any invertible matrix $R \in \mathbb{R}^{r \times r}$, the low-rank factors $(AR, R^{-1}B)$ yield the same adapter $AB$. In particular, if $(A, B)$ is a minimizer of the loss, then every $(AR, R^{-1}B)$ is also a minimizer, which yields a continuous manifold of minimizers. With that in mind, our novel theoretical analysis highlights that the conditioning of the loss can vary along this manifold: some minimizers are flatter than others, which, from an optimization perspective, makes them better candidates to converge to. Indeed, the loss around a flatter minimizer is

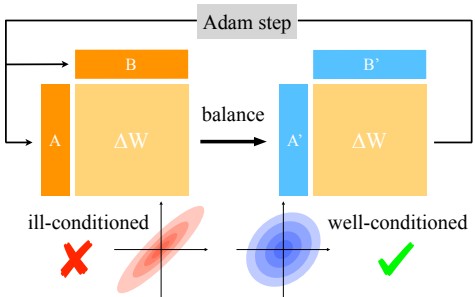

*Figure 1.* **BaLoRA in a nutshell.** BaLoRA projects the low-rank adapters $(A, B)$ on the balanced manifold after each optimizer step. This projection improves the conditioning of the loss while preserving the product $\Delta W = AB = A'B'$.

better conditioned, so that the asymptotic convergence rate to the minimizer is faster.

Our analysis identifies that *balanced minimizers*— minimizers $(A, B)$ satisfying $A^\top A = BB^\top$—achieve optimal conditioning of the loss. Although the balance condition has been studied in other contexts, such as linear networks (Nguegnang et al., 2024), matrix factorization (Ye & Du, 2021; Ghosh et al., 2025), and conservation laws in neural networks (Marcotte et al., 2023), *no previous work has connected it the conditioning of the loss* when performing low-rank adaptation. Similarly, while (Ghosh et al., 2025; Zhang et al., 2025) have introduced balanced parameterizations of the low-rank adapters, they do not show that such factorizations are optimal in terms of conditioning—which is one of the main contributions of this paper, as it provides a principled way to select the best-conditioned parameterization of a fixed adapter matrix. Leveraging these insights, we introduce BaLoRA, an extension of LoRA that enforces balance during training to improve conditioning and accelerate convergence. BaLoRA is computationally lightweight, theoretically grounded, and compatible with standard optimization pipelines. The code for our experiments is available at https://github.com/vcastin/balora. Our contributions are the following:

- We introduce **BaLoRA**, a novel PEFT method that enforces balanced low-rank adapters throughout optimization with negligible computational overhead. (Section 3)

- We theoretically analyze the conditioning of LoRA's limiting points when fine-tuning one layer of a deep, possibly non-linear network, proving that balanced minimizers exhibit optimal conditioning. Consequently, BaLoRA converges to a better-conditioned minimizer than LoRA, improving its asymptotic convergence rate. (Section 2)

- When optimizing with gradient descent, we demonstrate that BaLoRA iterations can be reformulated as an intrinsic

optimization scheme on the product $AB$, which provides an elegant and more interpretable geometric perspective on this algorithm. (Section 3)

- In our experiments on a range of large language models and datasets, BaLoRA consistently outperforms LoRA and matches or surpasses several state-of-the-art LoRA variants from the literature with negligible computational overhead. (Section 4)

### 1.1. Related Works

**Parameter-efficient fine-tuning.** To enable efficient adaptation without full retraining, residual adapters were first introduced for computer vision tasks (Rebuffi et al., 2017) and later extended to NLP through adapter-based transfer learning (Houlsby et al., 2019). Other prominent PEFT strategies include pruning-based approaches, such as Diff-Pruning (Guo et al., 2020), and low-rank adapters (Hu et al., 2022). LoRA and its variants have been widely adopted, ranging from bridging language models with non-language tasks via LIFT (Dinh et al., 2022) to fine-tuning image generation models (Fan et al., 2023). Theoretical analyses of LoRA in the Neural Tangent Kernel (NTK) regime have been conducted (Malladi et al., 2023; Jang et al., 2024), while its expressive power has been examined (Zeng & Lee, 2024).

**LoRA optimization.** LoRA is typically optimized using Adam (Kingma & Ba, 2017) or AdamW (Loshchilov & Hutter, 2017). Recent works have sought to adapt optimization strategies to low-rank structures. Riemannian approaches (Bogachev et al., 2025; Mo et al., 2025) tackle overparameterization through manifold-based optimization, but often require specialized solvers. Alternative algorithms for matrix factorization have been explored to strengthen convergence guarantees: Zhang & Fan (2024) analyze projected gradient descent and demonstrate convergence rates independent of condition numbers under specific assumptions; Ward & Kolda (2023) study alternating gradient descent, deriving bounds tied to spectral gaps; Zhang & Pilanci (2024) enhance gradient updates with Riemannian preconditioners; and Olikier et al. (2025) introduce Gauss–Southwell descent methods, emphasizing step-size and balancing interactions. Although matrix factorization shares similarities with LoRA, these studies do not address the fine-tuning of pretrained machine learning models.

**Initialization and convergence dynamics.** Standard LoRA initialization sets one low-rank matrix to zero and the other to Gaussian noise, ensuring the model initially retains the behavior of the pretrained model while enabling low-rank adaptations during training. The initial update $A_0 B_0$ is scaled by a factor $\alpha/r$, where $\alpha$ is a hyperparameter (Hu et al., 2022). Rank-stabilized scaling can be applied to mitigate gradient collapse at higher ranks (Kalajdzievski, 2023).

The convergence dynamics of LoRA are closely tied to results on deep linear networks and matrix factorization. For small step sizes, gradient descent converges under balancing conditions (Nguegnang et al., 2024), which become exact conservation laws of the gradient flow in the vanishing step size limit, thereby explaining implicit biases (Marcotte et al., 2023). Random initialization can guarantee global convergence in asymmetric low-rank matrix factorization (Ye & Du, 2021). Large step sizes, however, may push training toward the edge of stability (Cohen et al., 2021), a phenomenon extensively studied in linear networks (Ghosh et al., 2025; Chen & Bruna, 2023). For LoRA specifically, (Hayou et al., 2024b) propose assigning different learning rates to the low-rank factors to improve efficiency, while (Xu et al., 2025) analyze its dynamics through a gradient flow perspective, revealing an initial alignment phase followed by local convergence for small initialization scales. Finally, in the context of multi-layer perceptrons with ReLU activations, (Lebeurrier et al., 2026) leverage the rescaling symmetries of the network to reparametrize the weights at initialization, thus improving the conditioning of the training dynamics.

**Structural constraints.** Low-rank models are inherently overparameterized, but structural constraints can mitigate their inefficiencies. Orthogonality has been explored for optimization on the Stiefel manifold (Park et al., 2025; Lion et al., 2026) and in QR-based initialization (OLoRA (Büyükakyüz, 2024)). Other approaches exploit richer decompositions: DoRA (Liu et al., 2024) decomposes weights into magnitude and direction; butterfly-based orthogonal fine-tuning (BOFT (Liu et al., 2023b)) and Householder reflection adaptation (HRA (Yuan et al., 2024)) leverage structured orthogonal parameterizations; SVFT (Lingam et al., 2024) utilizes singular vectors of pretrained weights; VeRA (Kopiczko et al., 2023) reduces parameters by sharing low-rank random matrices with compact scaling vectors; GOAT (Fan et al., 2025) employs SVD-structured priors with mixture-of-experts alignment to refine initialization and scaling; and LoRA Done RITE (Yen et al., 2024) enforces invariance of the optimization process under scaling and rotation transformations of adapters.

## 2. Balanced Minimizers Are Best Conditioned

In this section, we analyze theoretically the asymptotic convergence rate of LoRA while fine-tuning one weight matrix of a general deep neural network (e.g., a Transformer (Vaswani et al., 2023)). We start by highlighting that the conditioning of the limit of LoRA is connected to the asymptotic convergence rate of the iterates. Denote $f(A, B) \in \mathbb{R}$ the loss to optimize. We assume throughout this section that $f$ takes the form of a regression loss $f(A, B) := \frac{1}{2}\|h(AB) - Z\|_F^2$, where $Z$ is the target matrix,

$\|\cdot\|_F$ is the Frobenius norm and $h(AB) \in \mathbb{R}^{d \times n}$ is the output of a generic, possibly non-linear neural network.

**Asymptotic rates and conditioning for gradient descent.** For simplicity, we first assume that LoRA is trained with gradient descent. LoRA iterations with step size $\gamma$ and initialization $(A_0, B_0)$ read

$$\begin{cases} A_{t+1} = A_t - \gamma\nabla_{A_t}f(A_t, B_t), \\ B_{t+1} = B_t - \gamma\nabla_{B_t}f(A_t, B_t). \end{cases} \quad (1)$$

When $f$ is the quadratic loss over a linear neural network, iterations (1) are known to converge to a minimizer $(A, B)$ of the loss (Nguegnang et al., 2024), with an unknown convergence rate. To gain insights on the convergence rate of $f$ to its optimum, we focus on the condition number $\kappa := \kappa(f)(A, B)$ of $f$ at a minimizer $(A, B)$. Letting $H$ be the Hessian of $f$ at $(A, B)$, it is defined as $\kappa := \lambda_{\max}(H)/\lambda_{\min \neq 0}(H)$, where $\lambda_{\min \neq 0}(H)$ is the smallest non-zero eigenvalue of $H$. The following classical result shows that a smaller condition number implies faster asymptotic convergence (e.g., Bach (2024)).

**Proposition 2.1.** *Assume the iterations (1) converge to a minimizer $(A, B)$ of $f$. Denoting $H$ the Hessian of $f$ at $(A, B)$, let $L := \lambda_{\max}(H)$ and $\mu := \lambda_{\min \neq 0}(H)$. Then,* $\limsup_{t \to +\infty} \frac{f(A_{t+1}, B_{t+1}) - f(A, B)}{f(A_t, B_t) - f(A, B)} \leq \max((1 - \gamma\mu)^2, (1 - \gamma L)^2)$. *Taking $\gamma = 2/(L + \mu)$ to minimize the right-hand side, and denoting $\kappa := L/\mu$,* $\limsup_{t \to +\infty} \frac{f(A_{t+1}, B_{t+1}) - f(A, B)}{f(A_t, B_t) - f(A, B)} \leq \left(\frac{\kappa - 1}{\kappa + 1}\right)^2$. *Hence, the smaller $\kappa \geq 1$, the faster the convergence to $(A, B)$ asymptotically.*

The intuition of Proposition 2.1 is to approximate the loss locally by the quadratic function given by its second-order Taylor's expansion around the minimizer of the trajectory. Hence, once the iterates are sufficiently close to their limit, the convergence rate in this regime (referred to as the asymptotic convergence rate) is governed by the conditioning of the Hessian of the loss at the limit.

**Asymptotic rates and conditioning for scaled sign-GD.** Proposition 2.1 is specific to gradient descent, while LoRA is typically trained with AdamW. Analyzing the local convergence rate of AdamW itself is still an open question, so we study instead a scaled sign-GD scheme, which corresponds to Adam without momentum (Pethick et al., 2025). Denoting $\theta_t := (A_t, B_t)$, we replace (1) by

$$\theta_{t+1} = \theta_t - \gamma\|\nabla f(\theta_t)\|_1 s_t, \qquad s_t \in \partial\|\nabla f(\theta_t)\|_1, \quad (2)$$

where $\partial$ denotes the subgradient. When all coordinates of $\nabla f(\theta_t)$ are non-zero, this simply becomes $\theta_{t+1} = \theta_t - \gamma\|\nabla f(\theta_t)\|_1 \text{sign}(\nabla f(\theta_t))$. The following proposition,

proved in Appendix C.1, shows that the local asymptotic rate of scaled sign-GD is controlled by the adapted $\ell^\infty$ condition number $\kappa_\infty$, which is itself bounded in terms of the usual Euclidean condition number $\kappa$.

**Proposition 2.2.** *Assume that the iterations* (2) *converge to a minimizer $\theta^\star$ of $f$, and denote by $H := \nabla^2 f(\theta^\star)$ the Hessian at that limit. Let $L_\infty := \sup_{u \neq 0} \langle Hu, u \rangle / \|u\|_\infty^2$ and $\mu_\infty := \inf_{u \in \ker(H)^\perp \setminus \{0\}} \|Hu\|_1^2 / \langle Hu, u \rangle$, and assume that $\mu_\infty > 0$. Define $\kappa_\infty := L_\infty / \mu_\infty$. If $D := r(a + b)$ denotes the ambient dimension of $(A, B)$, then $\kappa / D \leq \kappa_\infty \leq D\kappa$. Moreover, with the natural choice $\gamma = 1/L_\infty$, one has* $\limsup_{t \to +\infty} \frac{f(\theta_{t+1}) - f(\theta^\star)}{f(\theta_t) - f(\theta^\star)} \leq 1 - \frac{1}{\kappa_\infty} \leq 1 - \frac{1}{D\kappa}$.

In view of Propositions 2.1 and 2.2, it is therefore crucial to understand the condition number of the different minimizers of $f$. Losses of the form $f(A, B) = \frac{1}{2}\|h(AB) - Z\|_F^2$ have an $r^2$-dimensional manifold $\mathcal{M}$ of minimizers as, for any minimizer $(A, B)$, the couple $(AR, R^{-1}B)$ for $R$ an invertible $r \times r$ matrix leads to the same adapter $AB$, so is also a minimizer of $f$. Among such minimizers, we single out the submanifold $\mathcal{B}_{\min}$ of *balanced minimizers*, defined as all couples $(A, B) \in \mathcal{M}$ satisfying the *balancing condition* $A^\top A = BB^\top$. We prove in the next subsections that balanced minimizers are optimally conditioned, and discuss the balancing condition in Section 3.

## 2.1. The One-Layer Linear Case

We start with a tractable setting that still captures the complexity of the problem. Consider a pre-trained one-layer linear network $W \in \mathbb{R}^{a \times b}$ and a *target* $W^\star \in \mathbb{R}^{a \times b}$, representing the ideal fine-tuned model (Zeng & Lee, 2024). Let $Z := W^\star - W$ be the gap between the pretrained and target models. LoRA then consists in minimizing the loss

$$f : (A, B) \in \mathbb{R}^{a \times r} \times \mathbb{R}^{r \times b} \mapsto \frac{1}{2}\|Z - AB\|_F^2. \quad (3)$$

This setup generalizes matrix factorization (Ye & Du, 2021; Ghosh et al., 2025), where $\mathrm{rk}\, Z = r$, by allowing $Z$ to have rank $\mathrm{rk}\, Z > r$, which, to the best of our knowledge, has not been studied before. As detailed in the following paragraphs, having $\mathrm{rk}\, Z > r$ makes the mathematical analysis significantly more involved, as the Hessian of $f$—which has to be computed and diagonalized to investigate the conditioning of $f$—becomes the sum of two terms whose codiagonalization is non-trivial, while there is only one term in the matrix factorization case.

**The matrix factorization case.** Let us first assume that $\mathrm{rk}\, Z = r$. Then, the problem reduces to matrix factorization, and the minimal value of the loss $f$ is zero. We explicitly compute the Hessian of $f$ at any minimizer $(A, B)$, and determine its full spectrum and corresponding condition number.

**Proposition 2.3.** *Let $(A, B) \in \mathbb{R}^{a \times r} \times \mathbb{R}^{r \times b}$ be a global minimizer of the loss $f$ (3). Assume $\mathrm{rk}\, Z = r$. The Hessian of $f$ at $(A, B)$ reads*

$$H = \begin{pmatrix} (BB^\top) \otimes I_a & B \otimes A \\ B^\top \otimes A^\top & I_b \otimes (A^\top A) \end{pmatrix},$$

*and its eigenvalues are: 0 (with multiplicity $r^2$), $\sigma_i(A)^2 + \sigma_j(B)^2$ for $1 \leq i, j \leq r$ (each with multiplicity 1), $\sigma_i(A)^2$ (each with multiplicity $b - r$), and $\sigma_j(B)^2$ (each with multiplicity $a - r$). Therefore, $\kappa(f)(A, B) = (\sigma_1(A)^2 + \sigma_1(B)^2)/\min(\sigma_r(A)^2, \sigma_r(B)^2)$.*

Proposition 2.3, proved in Section A.1, establishes a direct link between the condition number of a minimizer $(A, B)$ and the singular values of $A$ and $B$, which allows us to identify optimally conditioned minimizers.

**Proposition 2.4.** *Assume $\mathrm{rk}\, Z = r$. Then, all balanced minimizers (i.e., such that $A^\top A = BB^\top$) have the minimal condition number $\kappa_{\min} = 2\sigma_1(Z)/\sigma_r(Z)$ among all minimizers.*

Combining Proposition 2.4 with Proposition 2.1 yields a closed-form connection between the best asymptotic convergence rate of the LoRA iterations and the spectrum of the target matrix $Z$. In particular, it shows that a target with a more spread-out spectrum corresponds to a more challenging matrix factorization problem. Furthermore, balanced minimizers achieve optimal conditioning, making them ideal limiting points for fast asymptotic convergence. These quantitative links between spectral structure, conditioning, and convergence rates offer novel insights into the matrix factorization problem.

**The general case.** We now investigate how our insights for matrix factorization extend to the general case $\mathrm{rk}, Z \geq r$. This scenario, where the adapters have a lower rank than the target, reflects the typical case of low-rank adaptation. Here, the residual $AB - Z \neq 0$ introduces additional off-diagonal terms in the Hessian.

**Proposition 2.5.** *Let $(A, B) \in \mathbb{R}^{a \times r} \times \mathbb{R}^{r \times b}$ be a global minimizer of the loss $f$ (3). Assume $\mathrm{rk}\, Z \geq r$. The Hessian of $f$ at $(A, B)$ reads*

$$H = \begin{pmatrix} (BB^\top) \otimes I_a & B \otimes A \\ B^\top \otimes A^\top & I_b \otimes (A^\top A) \end{pmatrix} +$$
$$\begin{pmatrix} 0 & (I_r \otimes (AB - Z))K_{r,b} \\ ((AB - Z)^\top \otimes I_r)K_{a,r} & 0 \end{pmatrix}$$

*where $K_{k,\ell}$ is the $k\ell \times k\ell$ matrix such that $\mathrm{vec}(X^\top) = K_{k,\ell}\mathrm{vec}(X)$ for any $X \in \mathbb{R}^{k \times \ell}$, with $\mathrm{vec}$ the vectorization operator.*

One can verify that $H$ is symmetric, since $(I_r \otimes (AB - Z))K_{r,b}(x \otimes y) = (I_r \otimes (AB - Z))(y \otimes x) = y \otimes (AB -$

$Z)x = K_{a,r}^\top((AB - Z)x \otimes y)$, as $K_{a,r}^\top = K_{r,a}$. The second term in $H$, that has positive and negative eigenvalues, makes the characterization of the conditioning of $H$ more challenging, especially to lower bound $\lambda_{\min\neq0}$. Below, we compute the sharpness of the Hessian at a minimizer and provide two bounds on its smallest eigenvalue. The proof is detailed in Section A.2.

**Proposition 2.6.** *Let $(A, B) \in \mathbb{R}^{a \times r} \times \mathbb{R}^{r \times b}$ be a global minimizer of the loss $f$ (3). The largest eigenvalue of the Hessian of $f$ at $(A, B)$ is $\lambda_{\max}(H) = \sigma_1(A)^2 + \sigma_1(B)^2$. Moreover, the smallest non-zero eigenvalue of $H$ satisfies,*

$$\lambda_{\min\neq0}(H) \geq \min(\sigma_r(A)^2, \sigma_r(B)^2) - \sigma_{r+1}(Z), \quad (4)$$

$$\lambda_{\min\neq0}(H) \leq \min(\sigma_r(A)^2, \sigma_r(B)^2). \quad (5)$$

*Finally, if $(A, B)$ is balanced, the lower bound (4) is maximized, equal to $\sigma_r(Z) - \sigma_{r+1}(Z)$, and becomes an equality: $\lambda_{\min\neq0}(H) = \sigma_r(Z) - \sigma_{r+1}(Z)$.*

Compared to matrix factorization, here balanced minimizers are still optimally conditioned, but the key quantity governing the intrinsic hardness of LoRA optimization shifts from $\sigma_r(Z)$ to the $r$-spectral gap $\sigma_r(Z) - \sigma_{r+1}(Z)$. This gap quantifies how well the rank-$r$ approximation separates from the discarded directions. The smaller the gap (and the larger $\sigma_1(Z)$), the slower the asymptotic convergence of the iterations (1) in the best case.

### 2.2. The Deep Non-Linear Case

The theoretical analysis in the previous section focuses on a simplified toy model (3). While a comprehensive analysis of Hessian conditioning for deeper architectures remains challenging, we can gain insights into why balancing improves conditioning in the case of fine-tuning a single-layer adapter in the interpolating regime (zero minimum loss).

Consider the regression loss $f(A, B) := \frac{1}{2}\|h(AB) - Z\|^2$, where $Z$ is the target matrix, $\|\cdot\|_F$ is the Frobenius norm and $h(AB) \in \mathbb{R}^{d \times n}$ is the output of a generic, possibly deep and non-linear neural network, with $n$ fixed inputs—$h$ is seen as a function of the LoRA adapter $AB$. For instance, for a 2-layer MLP with weights $(V, W)$ for which we fine-tune only the hidden-layer matrix $W$, one has $h(AB) := V \operatorname{ReLU}((W + AB)X)$, where $X \in \mathbb{R}^{b \times n}$ is the data matrix.

Assuming $nd \geq ab$ (a condition satisfied when sufficient data is available), the Jacobian $Dh(AB)$ is a rectangular and injective matrix. We define its conditioning as $\kappa(Dh(AB)) := \kappa(Dh(AB)^\top Dh(AB))^{1/2}$. The following Proposition, proved in Appendix A.3, provides a bound on the conditioning of the loss at a minimizer in the interpolation regime. It generalizes Proposition 2.3, which is recovered as a special case when $h = \mathrm{Id}$.

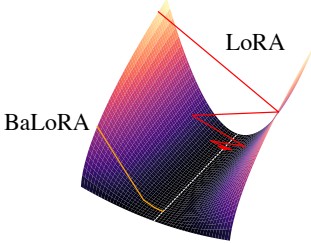

*Figure 2.* **The intuition behind BaLoRA.** By constraining the adapters to be balanced along the fine-tuning iterations, BaLoRA converges to balanced, and therefore, optimally conditioned, minimizer, reaching faster asymptotic convergence rates.

**Proposition 2.7.** *Let $(A, B)$ be a minimizer of the loss $f(A, B) := \frac{1}{2}\|h(AB) - Z\|^2$, such that $h(AB) = Z$. One has the following upper-bound on the conditioning of $f$ at $(A, B)$:*

$$\kappa(f)(A, B) \leq \kappa(Dh(AB))^2 \frac{\sigma_1(A)^2 + \sigma_1(B)^2}{\min(\sigma_r(A)^2, \sigma_r(B)^2)}.$$

*Moreover, taking $(A, B)$ to be balanced minimizes the upper-bound.*

Proposition 2.7 shows that balancing the singular values of $(A, B)$ minimizes the upper-bound on the conditioning at an interpolating point, which suggests an improved conditioning of the loss. Extending this analysis to the non-interpolating regime is an avenue for future work, and would bridge the gap with Proposition 2.6, which does not require interpolation but assumes $h = \mathrm{Id}$.

The results in this section suggest that steering the dynamics of (1) toward balanced adapters—whether through explicit or implicit regularization—can accelerate training in practice. Building on this insight, we propose a fine-tuning strategy that guides LoRA along balanced adapters, leading to faster convergence and improved stability in practice.

## 3. BaLoRA: Balanced Low-Rank Adaptation

Our results in Section 2 imply that balanced minimizers $(A, B) \in \mathcal{B}$ attain the optimal condition number, where $\mathcal{B} = \{(A, B) \mid A^\top A = BB^\top\}$ is the *balanced manifold* (Du et al., 2018). Building on this new insight, we thus propose to constrain LoRA iterations to stay on $\mathcal{B}$ by projecting the iterates after each optimizer (e.g., gradient or AdamW) step, to promote convergence to a better-conditioned minimizer with faster asymptotic rates (Figure 2).

As exposed in the following subsection, we introduce a submanifold $\mathcal{H} \subset \mathcal{B}$ of *hyperbalanced* matrices, which provides a more structured and efficient parameterization. We call Balanced Low-Rank Adaptation (BaLoRA) the novel fine-tuning method obtained by projecting to $\mathcal{H}$. Note that in the remainder, we use the term "manifold" with a slight

---

**Algorithm 1** Balanced projection

---

**Require:** $(A, B) \in \mathbb{R}^{a \times r} \times \mathbb{R}^{r \times b}$
  1: Compute polar decompositions $A = R_A S_A$ and $B = S_B R_B$
  2: Compute $S = S_A S_B \in \mathbb{R}^{r \times r}$
  3: Compute SVD decomposition $S = U \Sigma V^\top$
  4: **Return** $A^{\mathrm{proj}}, B^{\mathrm{proj}} = R_A(U\Sigma^{1/2}), (\Sigma^{1/2}V^\top)R_B$

---

abuse of language, since the sets we consider are manifolds with boundary.

### 3.1. Hyperbalanced Manifold and Balancing Map

Let $\mathbb{D}_+^r$ denote the set of $r \times r$ non-negative diagonal matrices with non-increasing diagonal values. We consider the submanifold (with boundary) $\mathcal{H} \subset \mathcal{B}$, which we call the *hyperbalanced manifold*: $\mathcal{H} = \{(A, B) \in \mathbb{R}^{a \times r} \times \mathbb{R}^{r \times b} \mid \exists S \in \mathbb{D}_+^r \text{ s.t. } A^\top A = BB^\top = S\}$. As detailed in Theorem B.1, one has the equivalent description of $\mathcal{H}$,

$$\mathcal{H} = \{(US^{1/2}, S^{1/2}V) \mid U^\top U = VV^\top = I_r, \ S \in \mathbb{D}_+^r\}. \tag{6}$$

This reformulation has two main consequences.

**Consequence 1: optimizing on $\mathcal{H}$ is equivalent to optimizing over low-rank matrices.** Denoting $\mathcal{N}_r$ the set of rank-$r$ matrices, (6) shows that $(A, B) \in \mathcal{H} \mapsto X := AB \in \mathcal{N}_r$ is surjective. For a function $g(X)$, define $f(A, B) := g(AB)$. Then, $\min_{X \in \mathcal{N}_r} g(X)$ and $\min_{(A,B) \in \mathcal{H}} f(A, B)$ are equivalent problems. Working with variables $(A, B) \in \mathcal{H}$ therefore provides a computationally convenient framework for low-rank optimization, while also taking advantage of the improved conditioning discussed in the previous section.

**Consequence 2: definition of the balancing map $P$ "projecting" onto $\mathcal{H}$.** Given $(A, B)$ with $X = AB$, take any reduced SVD with decreasing singular values $X = U S V^\top$. The *balancing map* is defined as

$$P(A, B) := (US^{1/2}, S^{1/2}V^\top). \tag{7}$$

The reformulation in (6) shows that $P$ "projects" onto $\mathcal{H}$. Although this is not an orthogonal projector, Theorem B.3 in the appendix shows that it exhibits a "projection-like" behavior, namely, it defines a smooth retraction (Absil & Malick, 2012) onto $\mathcal{H}$. Consequently, results from Riemannian optimization (Boumal, 2023) can be applied to analyze the convergence of the BaLoRA-GD method, *i.e.*, gradient descent combined with $P$ to keep the iterates on $\mathcal{H}$, as detailed in the next section. Another important property of the balancing map $P$ is that it preserves the product, unlike the orthogonal projector: denoting $(\tilde{A}, \tilde{B}) := P(A, B)$, then $\tilde{A}\tilde{B} = AB$. This preservation guarantees that the loss remains *unchanged*, which is essential for the intrinsic reformulation described in Section 3.2. The procedure to

---

**Algorithm 2** BaLoRA training

---

**Require:** Pretrained weights, to be finetuned: $W_\ell \in \mathbb{R}^{a \times b}$ for $1 \le \ell \le L$; rank $r$; learning rate $\eta$
  1: **Initialize:** $A_\ell^0 = 0 \in \mathbb{R}^{a \times r}$ and $B_\ell^0 \in \mathbb{R}^{r \times b}$ using Kaiming initialization, for $1 \le \ell \le L$
  2: **for** $t = 0, 1, \ldots, T-1$ **do**
  3:     Sample a mini-batch of data
  4:     Compute corresponding gradients $\nabla_{A_\ell^t} f$ and $\nabla_{B_\ell^t} f$ for the LoRA loss $f(A_1^t, B_1^t, \ldots, A_L^t, B_L^t)$
  5:     **Optimizer Step:** Update low-rank factors
  6:     $(\tilde{A}_\ell^{t+1}, \tilde{B}_\ell^{t+1}) \leftarrow \mathrm{Step}(A_\ell^t, B_\ell^t, \nabla_{A_\ell^t} f, \nabla_{B_\ell^t} f, \eta)$
  7:     **Balanced Projection:**
  8:     $(A_\ell^{t+1}, B_\ell^{t+1}) \leftarrow P(\tilde{A}_\ell^{t+1}, \tilde{B}_\ell^{t+1})$
  9: **end for**
 10: **Return** Final adapters $(A_1^T, B_1^T), \ldots, (A_L^T, B_L^T)$

---

efficiently compute $P$ is given in Algorithm 1, and has computational complexity $\mathcal{O}((a + b)r^2)$ for $a, b \gg r$, which adds negligible overhead to the cost of the optimizer step (see Section 4).

### 3.2. BaLoRA and BaLoRA-GD

> **Definition.** The *BaLoRA* method consists in applying $P(A, B) = A^{\mathrm{proj}}, B^{\mathrm{proj}}$ at the end of each step of an optimization scheme, as defined in Algorithms 1 and 2. When combined with Adam, we simply refer to the resulting algorithm as *BaLoRA*. When applied to the iterates of gradient descent, we call it *BaLoRA-GD*.

While BaLoRA (with Adam) is a heuristic method whose full rigorous theoretical analysis is beyond the scope of this work, we show that the gradient descent variant, BaLoRA-GD, exhibits a striking intrinsic behavior. Recall that BaLoRA-GD iterates for $k \ge 0$ and some stepsize $\tau_k > 0$, starting from any initialization $(A_0, B_0)$, read $(A_{k+1}, B_{k+1}) = P(A_k - \tau_k \nabla_A f(A_k, B_k), B_k - \tau_k \nabla_B f(A_k, B_k))$.

**Intrinsic BaLoRA-GD.** Consider a loss function $f(A, B) = g(X)$, where $X = AB$ and $g: \mathbb{R}^{a \times b} \to \mathbb{R}$ is smooth. This general setting encompasses all LoRA losses. As shown in Proposition 3.1 below, proved in Appendix B.1, the BaLoRA-GD iteration can be written entirely as an intrinsic gradient descent on the manifold of rank-$r$ matrices $\mathcal{N}_r$ endowed with a Riemannian metric. For $X \in \mathcal{N}_r$, the inverse of this metric is given by the symmetric positive (semi)definite linear operator $H_X[W] := (XX^\top)^{1/2}W + W(X^\top X)^{1/2}$. When restricted to symmetric positive definite (SPD) matrices, this operator reduces to $H_X[W] = XW + WX$ and coincides with the inverse of the Bures metric. This metric is well known in optimal transport on Gaussian measures

and provides a natural tool for optimization over the cone of SPD matrices (Bhatia et al., 2019). By abuse of terminology, we will refer to $H_X$ as the inverse Bures metric on the larger manifold $\mathcal{N}_r$.

**Proposition 3.1** (Intrinsic update on $X = AB$). *Let* $f(A, B) = g(AB)$. *Denote by* $(A_k, B_k)$ *the BaLoRA-GD iterates and set* $X_k := A_k B_k$. *Then for* $k \geq 1$,

$$X_{k+1} = R\big(X_k, -\tau_k \Delta_k\big), \tag{8}$$

$$\text{where } R(X, \delta) = X + \delta - H_X^{-1}[\delta]\, X^\top\, H_X^{-1}[\delta], \tag{9}$$

*and* $\Delta_k := H_{X_k}[\nabla g(X_k)]$ *is the Riemannian gradient associated with the Bures metric, $R$ is a retraction on $\mathcal{N}_r$.*

Note that although $H_X^{-1}[\delta]$ is not uniquely defined when $X$ is rank-deficient, the quantity $R(X, \delta)$ is uniquely defined. Moreover, (8) may fail for $k = 0$ if $(A_0, B_0)$ does not belong to the balancing set $\mathcal{H}$.

Equation (8) is the canonical way to express a Riemannian gradient descent on a manifold using a retraction (Boumal, 2023). When $\tau_k \to 0$, this iteration converges to the gradient flow $\dot{X} = -H_X[\nabla g(X)]$. BaLoRA-GD can therefore be interpreted as an efficient implementation of gradient descent with respect to the Bures metric, leveraging computations on the factored variables $(A, B)$ instead of working directly with $X$.

**BaLoRA initialization.** Throughout the paper, we initialize BaLoRA in the same way as standard LoRA, with $A = 0$ and $B$ following the Kaiming initialization. However, the BaLoRA method is compatible with a variety of different initializations, such as the OLoRA (Büyükakyüz, 2024) or the LoRA-GA (Wang et al., 2024) initialization. Investigating whether some of these initializations are particularly suited to BaLoRA is an interesting avenue for future work.

# 4. Experiments

We present an empirical evaluation of BaLoRA, demonstrating improved performance and convergence speed with negligible computational overhead. Our experiments cover fine-tuning tasks on synthetic and real-world data across multiple pre-trained architectures, complemented by ablation studies highlighting BaLoRA's robustness to hyperparameter choice and adapter rank.

## 4.1. Synthetic Experiments

We first compare the dynamics of BaLoRA and standard LoRA on a toy framework which aligns closely with our theoretical analysis. Specifically, we consider the optimization problems, $\min_{(A,B)} \|W^\star - (W + AB)\|_F^2$ and $\min_{(A,B)} \|W^\star - (W_1 + A_1 B_1)(W_2 + A_2 B_2)\|_F^2$ for $(A, B) \in \mathbb{R}^{a \times r} \times \mathbb{R}^{r \times a}$. The first problem is encompassed by the setting studied in Section 2, while the second extends

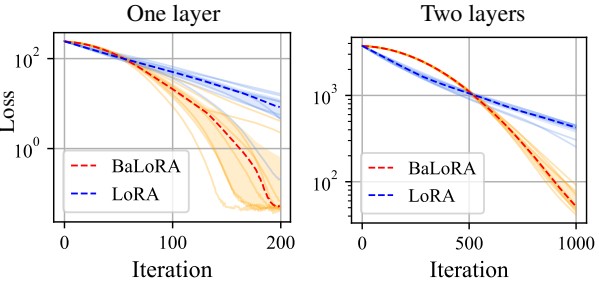

*Figure 3.* **Synthetic experiments.** Evolution of the loss of LoRA vs. BaLoRA. The dotted lines are the median of 8 curves with different seeds for the initialization, for a fixed target. Both methods use the standard LoRA init, with $A_0 = 0$, $B_0$ random Gaussian, a scaling $\alpha/r = 1$, and a LoRA rank of 4. The left (resp. right) plot corresponds to a square one-layer linear network of size 20 (resp. a two-layer linear network of size 20, whose layers are both fine-tuned). After a slower start, BaLoRA converges faster in both situations.

this setting to fine-tuning simultaneously both layers of a 2-layer linear network, which introduces more complex interactions between layers. We apply LoRA and BaLoRA, optimized with Adam, to these problems across eight different initialization seeds. In Figure 3, we report the loss over iterations for a fixed learning rate. We see that for both configurations (one or two layers), BaLoRA starts slower than LoRA, then enters a fast convergence regime where it significantly outperforms LoRA. This confirms our insights from Section 2 and extends their scope to fine-tuning two layers simultaneously.

## 4.2. Experiments with Large Language Models

We scale up the experiments to large language models and real-world data. We fine-tune the pretrained models Llama-3.2-3B (Meta AI, 2024) and Qwen-2.5-3B (Qwen et al., 2025), evaluating their abilities in language modeling with the dataset Wikitext-2-raw-v1 (Merity et al., 2016), in dialogue with the dataset WizardLM (Xu et al., 2023), instruction-following using Alpaca (Taori et al., 2023a) and OpenOrca (Lian et al., 2023), their mathematical reasoning with the datasets MetaMathQA (Yu et al., 2023), GSM8K (Cobbe et al., 2021) and DeepMind Mathematics (Saxton et al., 2019), their coding abilities with the dataset Code-Feedback (Zheng et al., 2025), their scientific reasoning and text processing with arXiv (arXiv.org submitters, 2024), and their general natural language understanding with OpenHermes (Teknium, 2023). Since our focus is on optimization speed, we compare methods by reporting the loss on held-out test sets.

**General setup.** In all the experiments, we simultaneously fine-tune all MLP layers with the optimizer AdamW (Loshchilov & Hutter, 2017), while keeping the atten-

tion layers frozen. The learning rate remains constant throughout the training. We perform extensive experiments with a LoRA rank equal to 8, complemented with a rank ablation ($r \in \{2, 4, 8, 16, 32, 64, 128\}$) on the datasets DeepMind Mathematics and arXiv. For the datasets Wikitext, MetaMathQA and GSM8K, we tune the learning-rate and the scaling at initialization—we call scaling the scalar $\alpha$ that multiplies the LoRA adapter: $W + \alpha AB$—independently for each method, by running a sweep of learning rates $\gamma$ and scalings $\alpha$, and selecting the pair $(\gamma, \alpha)$ with the smallest evaluation loss at the end of the fine-tuning. For every other dataset $\mathcal{D} \in \{$Alpaca, CodeFeedback, OpenHermes, OpenOrca, WizardLM, DeepMind Mathematics, arXiv$\}$, we use the MetaMathQA sweep as a reference: we pick the learning rate and scaling that have the smallest evaluation loss after fine-tuning on a subset of MetaMathQA of size close to $\mathcal{D}$, which provides dataset-specific and method-specific hyperparameters without having to run an entire sweep.

**Baselines.** We compare BaLoRA with several related LoRA-variants. We do not include other PEFT methods such as Galore, as they are not directly comparable to the adaptation budget of LoRA-style methods.

1. *Standard LoRA* (Hu et al., 2022): each pretrained weight matrix is fine-tuned by adding a trainable product of low-rank adapters $AB$. $B$ is initialized with Kaiming initialization and $A$ is initially set to 0.

2. *LoRA-GA* (Wang et al., 2024): the adapters $A, B$ are initially balanced and such that $AB$ is the best rank-$r$ approximation of the full gradient of the loss (as a function of the weights, without LoRA adapters). Then, $A, B$ are optimized without constraints, as in LoRA. In particular, $(A, B)$ does not stay balanced.

3. *OLoRA* (Büyükakyüz, 2024): $A, B$ are initialized as the QR decomposition of the pretrained weight matrix, truncated at rank $r$. In particular, $A, B$ are not balanced and are then optimized as in LoRA.

4. *DoRA* (Liu et al., 2024): the structure of the low-rank adaptation is changed by decoupling the magnitude of $AB$ and its direction. Both components are learnable.

5. *RefLoRA* (Zhang et al., 2025): like BaLoRA, this method enforces balancedness of the adapters $(A, B)$ across the optimization. However, the balancing is obtained by applying the map $\tilde{P}(A, B) = (AS^{1/2}, S^{-1/2}B)$ where $S = (A^\top A)^{-1/2}[(A^\top A)^{1/2}(BB^\top)(A^\top A)^{1/2}]^{1/2}(A^\top A)^{-1/2}$, which leads to a different balanced pair as with the BaLoRA projection $P(A, B)$. Moreover, RefLoRA starts the optimization with 100 steps of standard LoRA.

6. *Lora-RITE* (Yen et al., 2024): the AdamW optimizer is replaced by a transformation-invariant optimizer, using adaptive matrix preconditioning.

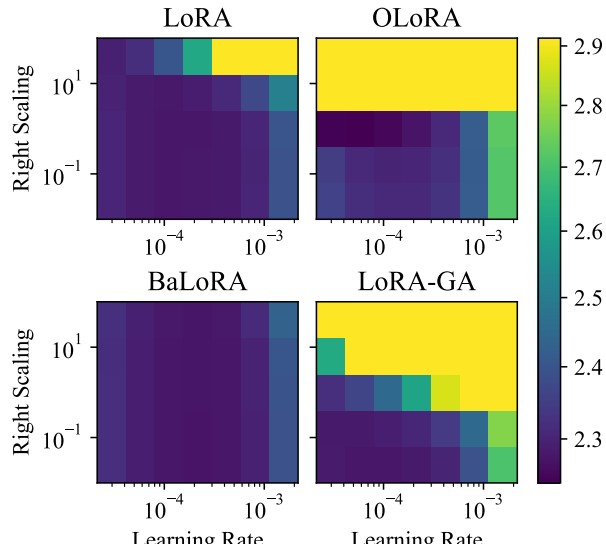

*Figure 4.* Hyperparameter sensitivity analysis (learning rates, initialization scalings) when fine-tuning Llama-3.2-3B on Wikitext-2-raw-v1. We observe that BaLoRA is significantly more stable to high scalings than all methods, and more stable to high learning rates than OLoRA and LoRA-GA.

7. *LORO* (Mo et al., 2025): at each step, each pair of adapters is jointly updated to ensure the product is transported along the Riemannian gradient of the manifold of rank-$r$ matrices.

**Wikitext.** We fine-tune Llama-3.2-3B (Meta AI, 2024) and Qwen-2.5-3B (Qwen et al., 2025) on the train split of Wikitext-2-raw-v1 (Merity et al., 2016) (36.7k samples), with a context length of 1024 tokens. Learning rate and initialization scaling are selected from a $10 \times 10$ sweep, and performance is evaluated via cross-entropy loss on the test split (4.36k samples). Test losses are reported in Table 4. Figure 7 shows the test loss as a function of runtime for Llama, accounting for the computational overhead of each variant: under this metric, BaLoRA stands out as the best method. Figures 4 and 8 compare hyperparameter sensitivity for Llama, highlighting that BaLoRA performs well across a wider range of learning rates and initialization scales.

**MetaMathQA.** We fine-tune We fine-tune Qwen-2.5-3B on a 100k-sample subset of MetaMathQA using a $10 \times 10$ sweep over learning rate $\gamma$ and scaling $\alpha$, averaging losses over 3 runs (Table 5). This sweep then serves as a reference for hyperparameter selection on new datasets: we record the evaluation loss for each $(\gamma, \alpha)$ pair every 0.03 epoch fractions, and for a new dataset $\mathcal{D}$, we find the epoch fraction $k$ such that $k \times 0.03 \times 10^5$ is closest to $|\mathcal{D}|$, then select the pair minimizing the MetaMathQA eval loss at fraction $k$.

**Alpaca, CodeFeedback, OpenHermes, OpenOrca, WizardLM.** We fine-tune Qwen-2.5-3B with BaLoRA against several baselines on the datasets Alpaca, CodeFeedback,

*Table 1.* Final evaluation loss of BaLoRA versus several LoRA variants, when fine-tuning Qwen-2.5-3B over several datasets (A=Alpaca, CF=CodeFeedback, OH=OpenHermes, OO=OpenOrca, WLM=WizardLM) with $r = 8$. Best test loss is in bold, second best is underlined. BaLoRA and RefLoRA, which impose balanced iterations, outperform the other methods.

| Method/Dataset | A | CF | OH | OO | WLM |
|---|---|---|---|---|---|
| LoRA | 1.352 | 0.638 | 0.707 | 0.774 | 0.663 |
| DoRA | 1.352 | 0.639 | 0.707 | 0.776 | 0.662 |
| LoRA-RITE | 1.353 | 0.639 | 0.707 | 0.776 | 0.663 |
| LoRO | 1.504 | 0.669 | 0.750 | 0.859 | 0.689 |
| OLoRA | 1.360 | 0.641 | 0.712 | 0.782 | 0.666 |
| RefLoRA | 1.350 | **0.638** | **0.706** | **0.773** | **0.661** |
| BaLoRA | **1.350** | 0.638 | 0.707 | 0.773 | 0.662 |

OpenHermes, OpenOrca and WizardLM. The rank is set to 8. The learning rate and the initial scaling are chosen optimally per-method and per-dataset from the MetaMathQA sweep, as explained above. Results are reported in Table 1 and Figure 10. BaLoRA consistently outperforms LoRA and ranks in the top 2 across all settings. The best-performing method is RefLoRA, which enforces balanced iterates during optimization — notably, the two balanced methods outperform all others, supporting the claim that balanced iterates accelerate convergence. A comparison of BaLoRA and RefLoRA is provided in Appendix C.

**Impact of the rank: DeepMind Mathematics and arXiv.** We test the impact of the rank of the adapters by fine-tuning Qwen-2.5-3B on 1B-token subsets of DM Mathematics and arXiv, with BaLoRA versus several baselines, for $r \in \{2, 4, 8, 16, 32, 64, 128\}$. For a given method, we take the same learning rate and initial scaling for all rank values, selected from the MetaMathQA sweep with $r = 8$. The results are reported in Tables 2 and 8, and Figures 5 and 11. BaLoRA significantly outperforms the other methods, including RefLoRA, for higher ranks ($r \in \{64, 128\}$).

**Peak GPU memory comparison.** We compare the memory consumption of several benchmarked methods, including BaLoRA. According to the results reported in Table 6, BaLoRA adds negligible memory overhead to standard LoRA.

**Discussion.** In our experiments, BaLoRA and RefLoRA consistently stand out as the best methods. Both enforce balanced iterates throughout optimization, supporting the claim that balancing accelerates convergence. Additionally, BaLoRA excels at fine-tuning with large ranks while adding negligible computational overhead over standard LoRA.

## 5. Conclusion

This paper presents a theoretical analysis of LoRA's convergence dynamics, revealing how its overparameterization

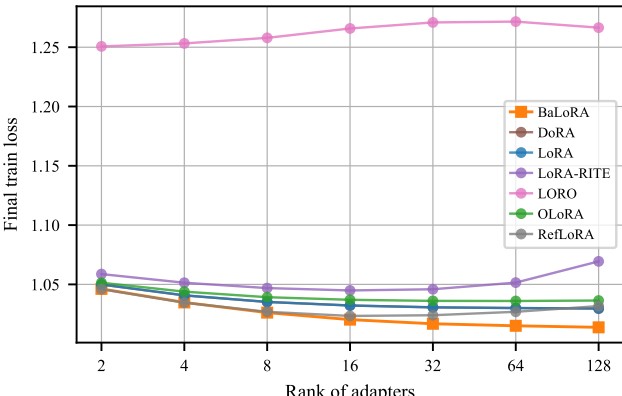

*Figure 5.* Impact of the rank of the adapters on the final train loss when fine-tuning Qwen-2.5-3B on a 1B subset of the DeepMind Mathematics Dataset, selecting per-method optimal hyperparameters from the MetaMathQA sweep with $r = 8$. BaLoRA is the best method for almost all ranks, and has a clear edge for larger ranks.

*Table 2.* Impact of the rank of the adapters on the final train loss when fine-tuning Qwen-2.5-3B on a 1B subset of the DeepMind Mathematics Dataset. For each method, the learning rate and the scaling at initialization are selected from a sweep on MetaMathQA: concretely, we pick the hyperparameters that achieve the smallest evaluation loss when fine-tuning Qwen on MetaMathQA with $r = 8$, which provides a fair procedure for choosing the hyperparameters of each method. Best loss is in bold, second best loss underlined. BaLoRA is the best method for almost all ranks, and has a clear edge for larger ranks. The full table is in Appendix 7.

| Method/rank | 8 | 16 | 32 | 64 | 128 |
|---|---|---|---|---|---|
| LoRA | 1.035 | 1.032 | 1.031 | 1.030 | 1.030 |
| DoRA | 1.035 | 1.032 | 1.031 | 1.030 | 1.030 |
| LoRA-RITE | 1.047 | 1.045 | 1.046 | 1.052 | 1.069 |
| LoRO | 1.258 | 1.266 | 1.271 | 1.272 | 1.267 |
| OLoRA | 1.039 | 1.037 | 1.036 | 1.036 | 1.036 |
| RefLoRA | 1.027 | 1.023 | 1.024 | 1.027 | 1.032 |
| BaLoRA | **1.026** | **1.020** | **1.017** | **1.015** | **1.014** |

induces varying condition numbers across minimizers. We identify balanced minimizers — which achieve optimal conditioning — as a critical factor for efficient optimization. Building on this insight, we introduce BaLoRA, a novel extension of LoRA that enforces balance by projecting adapters onto the hyperbalanced manifold after each optimization step. This preserves the adapted weight matrix while improving its conditioning, yielding faster convergence and greater robustness to hyperparameter choices, with negligible overhead. Our empirical evaluations on Llama-3.2-3B and Qwen-2.5-3B show that BaLoRA consistently outperforms standard LoRA and matches or surpasses state-of-the-art variants in accuracy and stability across learning rates, initialization scales, and adapter ranks — with a particular advantage at large ranks ($r \in \{64, 128\}$).

## Impact Statement

This paper presents work whose goal is to advance the field of Machine Learning. There are many potential societal consequences of our work, none which we feel must be specifically highlighted here.

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

# A. Spectrum of the limiting Hessian

## A.1. Proof of Proposition 2.3

Computing $H$ is straightforward. To diagonalize this matrix, notice that $H = MM^\top$ with

$$M := \begin{pmatrix} B \otimes I_a \\ I_b \otimes A^\top \end{pmatrix} \in \mathbb{R}^{r(a+b) \times ab}.$$

- **Kernel of $H$.** The kernel of $H$ is equal to the kernel of $M^\top$, which can be written as $\{\mathrm{vect}\,(AR, -RB) : R \in \mathbb{R}^{r \times r}\}$. Indeed, unvectorizing the equation $M^\top \mathrm{vect}\,(D, E) = 0$ gives $DB + AE = 0$, and such $(D, E)$ can be rewritten as $(AR, -RB)$ for $R = A^+ D = -EB^+$. Therefore, $\ker H$ is of dimension $r^2$.

- **Non-zero spectrum of $H$.** To find the non-zero eigenvalues of $H$ and associated eigenvectors, we use the following observation.

  **Lemma A.1.** *Let $M$ be a matrix and $x$ be an eigenvector of $M^\top M$ associated with a non-zero eigenvalue $\lambda$. Then $Mx$ is an eigenvector of $MM^\top$, associated with the eigenvalue $\lambda$.*

  We have

  $$M^\top M = (B^\top B) \otimes I_a + I_b \otimes (AA^\top).$$

  Let $(\lambda, x)$ be an eigenpair of $AA^\top$ and $(\mu, y)$ an eigenpair of $B^\top B$. Then $(\lambda + \mu, y \otimes x)$ is an eigenpair of $M^\top M$. Therefore, denoting $\lambda_1, \dots, \lambda_r$ and $\mu_1, \dots, \mu_r$ the non-zero eigenvalues of $AA^\top$ and $B^\top B$ respectively, with associated unit eigenvectors $x_1, \dots, x_r$ and $y_1, \dots, y_r$, and denoting $x_{r+1}, \dots, x_a$ and $y_{r+1}, \dots, y_b$ unit bases of $\ker AA^\top$ and $\ker B^\top B$, the eigenpairs of $M^\top M$ associated with non-zero eigenvalues are

  $$(\lambda_i + \mu_j, y_j \otimes x_i)_{1 \le i,j \le r} \cup (\lambda_i, y_{r+j} \otimes x_i)_{\substack{1 \le j \le b-r \\ 1 \le i \le r}} \cup (\mu_j, x_j \otimes x_{r+i})_{\substack{1 \le i \le a-r \\ 1 \le j \le r}}.$$

  Using Lemma A.1, the eigenpairs of $MM^\top$ with non-zero eigenvalues are thus

  $$(\lambda_i + \mu_j, M(y_j \otimes x_i))_{1 \le i,j \le r} \cup (\lambda_i, M(y_{r+j} \otimes x_i))_{\substack{1 \le j \le b-r \\ 1 \le i \le r}} \cup (\mu_j, M(y_j \otimes x_{r+i}))_{\substack{1 \le i \le a-r \\ 1 \le j \le r}}.$$

## A.2. Proof of Proposition 2.6

Sharpness of the Hessian. Let us first compute the largest eigenvalue of $H$. Denote $H_1 := \begin{pmatrix} (BB^\top) \otimes I_a & B \otimes A \\ B^\top \otimes A^\top & I_b \otimes (A^\top A) \end{pmatrix}$ and $H_2 := \begin{pmatrix} 0_{ar \times ar} & (I_r \otimes (AB-Z))K_{r,b} \\ ((AB-Z)^\top \otimes I_r)K_{a,r} & 0_{br \times br} \end{pmatrix}$, so that $H = H_1 + H_2$.

**Lemma A.2.** *The largest eigenvalue of $H_2$ is $\sigma_{r+1}(Z)$. The smallest eigenvalue of $H_2$ is $-\sigma_{r+1}(Z)$.*

*Proof.* Denote $G := (I_r \otimes (AB-Z))K_{r,b} = (((AB-Z)^\top \otimes I_r)K_{a,r})^\top \in \mathbb{R}^{ra \times rb}$. Let $(u_1, \dots, u_{ra})$ and $(v_1, \dots, v_{rb})$ be respectively the left and right eigenvectors of $G$. Denote $\tau_1 \ge \dots \ge \tau_\varrho$ the singular values of $G$, with $\varrho := \mathrm{rk}\, G = r(\min(a, b) - r)$.

- The Kernel of $H_2$ is the span of $\{\begin{pmatrix} u_j \\ 0 \end{pmatrix} : j = \varrho + 1, \dots, ar\} \cup \{\begin{pmatrix} 0 \\ v_k \end{pmatrix} : k = \varrho + 1, \dots, br\}$.

- For $i = 1, \dots, \varrho$, let $x_i^+ := \begin{pmatrix} u_i \\ v_i \end{pmatrix}$ and $x_i^- := \begin{pmatrix} u_i \\ -v_i \end{pmatrix}$. Then $x_i^+$ (resp. $x_i^-$) is an eigenvector of $H_2$ associated with the eigenvalue $\sigma_i$ (resp. $-\sigma_i$). The $\sigma_i$ can be easily computed, they are of the form $-\sigma_k(Z)$ for $k = r + 1, \dots, \min(a, b)$, which proves the result.

$\square$

The eigenvectors of $H_1$ form a basis of the space $\mathbb{R}^{(a+b)r}$. We will prove that for any eigenvector $u$ of this basis, it holds $|Hu| \le (\sigma_1(A)^2 + \sigma_1(B)^2)|u|$.

- If $u$ is in the kernel of $H_1$ and has unit norm, then $|Hu| = |H_2u| \leq \|H_2\|_2 = \sigma_{r+1}(Z) \leq \sigma_1(Z) \leq \sigma_1(A)\sigma_1(B) \leq \sigma_1(A)^2 + \sigma_1(B)^2$.

- If $u$ is of the form $M(y_j \otimes x_i)$ with the notations of the proof of Proposition 2.3, then

$$H_2u = \begin{pmatrix} (I_r \otimes (AB - Z))(A^\top x_i \otimes y_j) \\ ((AB - Z)^\top \otimes I_r)(x_i \otimes By_j) \end{pmatrix} = 0, \tag{10}$$

so $|Hu| = |H_1u| \leq (\sigma_1(A)^2 + \sigma_1(B)^2)|u|$.

- If $u$ is of the form $M(y_{r+j} \otimes x_i)$ for $1 \leq i \leq r$ and $1 \leq j \leq b-r$, it is easy to check that $H_1u$ and $H_2u$ are orthogonal. Then:

$$\begin{aligned} |Hu| &= \sqrt{|H_1u|^2 + |H_2u|^2} \\ &\leq \sqrt{\sigma_i(A)^4|u|^2 + \sigma_{r+1}(Z)^2|u|^2} \\ &= \sqrt{\sigma_i(A)^4 + \sigma_{r+1}(Z)^2}|u|. \end{aligned}$$

We therefore need to prove that $\sqrt{\sigma_i(A)^4 + \sigma_{r+1}(Z)^2} \leq \sigma_1(A)^2 + \sigma_1(B)^2$, or equivalently that $\sigma_1(A)^2 + \sigma_1(B)^2 - \sqrt{\sigma_1(A)^4 + \sigma_{r+1}(Z)^2} \geq 0$. We have $\sigma_1(B)^2 \geq \sigma_1(Z)^2/\sigma_1(A)^2$. Then

$$\begin{aligned} \sigma_1(A)^2 &+ \sigma_1(B)^2 - \sqrt{\sigma_1(A)^4 + \sigma_{r+1}(Z)^2} \\ &\geq \sigma_1(A)^2 + \sigma_1(Z)^2/\sigma_1(A)^2 - \sqrt{\sigma_1(A)^4 + \sigma_{r+1}(Z)^2} \\ &\geq \sigma_1(A)^2 + \sigma_{r+1}(Z)^2/\sigma_1(A)^2 - \sqrt{\sigma_1(A)^4 + \sigma_{r+1}(Z)^2} \\ &= (\sigma_1(A)^4 + \sigma_{r+1}(Z)^2)\left(\frac{1}{\sigma_1(A)^2} - \frac{1}{\sqrt{\sigma_1(A)^4 + \sigma_{r+1}(Z)^2}}\right) \\ &\geq 0, \end{aligned}$$

which proves the result.

Smallest non-zero eigenvalue of the Hessian. We have $\lambda_{\min \neq 0}(H) = \lambda_{r(a+b)-r^2}(H)$, as the Kernel of $H$ has dimension $r^2$. Equation 10 shows that all the eigenvalues of $H_1$ are also eigenvalues of $H_2$. Therefore, $\lambda_{r(a+b)-r^2}(H) \leq \min(\sigma_r(A)^2, \sigma_r(B)^2)$. For the lower bound, we apply the Weyl inequality:

$$\begin{aligned} \lambda_{r(a+b)-r^2}(H) &= \lambda_{r(a+b)-r^2}(H_1 + H_2) \\ &\geq \lambda_{r(a+b)-r^2}(H_1) + \lambda_{r(a+b)}(H_2) \\ &= \min(\sigma_r(A)^2, \sigma_r(B)^2) - \sigma_{r+1}(Z), \end{aligned}$$

according to Lemma A.2.

When the minimizer is balanced, it holds $\sigma_r(A)^2 = \sigma_r(B)^2 = \sigma_r(Z)$. This is the maximal value for the lower bound. Indeed, let $(A, B)$ be any minimizer of the loss $f$, i.e., $AB = LR_r(Z)$. Denote $U\Sigma V^\top$ the thin SVD of $LR_r(Z)$, i.e., with $\Sigma \succ 0$ of size $r \times r$. We can write $A = U\Sigma^{1/2}P$, $B = P^{-1}\Sigma^{1/2}V^\top$ for some invertible matrix $P \in GL_r(\mathbb{R})$. Then $\sigma_r(Z) = \sigma_r(U\Sigma V^\top) = \sigma_r(\Sigma^{1/2}PP^{-1}\Sigma^{1/2}) \geq \sigma_r(\Sigma^{1/2}P)\sigma_r(P^{-1}\Sigma^{1/2})$ by the Weyl inequality. Hence, $\sigma_r(Z) \geq \sigma_r(A)\sigma_r(B)$. We have proven that balanced minimizers maximize the lower bound $\min(\sigma_r(A)^2, \sigma_r(B)^2) - \sigma_{r+1}(Z)$.

Now, it is easy to check that when $(A, B)$ is balanced, the vector $\begin{pmatrix} o_r \otimes u_{r+1} \\ v_{r+1} \otimes o_r \end{pmatrix}$ with

- $o_r$ an eigenvector of $A^\top A = BB^\top$ associated with eigenvalue $\sigma_r(Z)$,

- $u_{r+1}$ the column $r + 1$ of $U$,

- $v_{r+1}$ the column $r + 1$ of $V$,

is an eigenvector of $H$ with the eigenvalue $\sigma_r(Z) - \sigma_{r+1}(Z)$, which proves that $\lambda_{\min \neq 0}(H) = \sigma_r(Z) - \sigma_{r+1}(Z)$ when $(A, B)$ is balanced.

## A.3. Proof of Proposition 2.7

We denote $\theta := (A, B)$ and $\phi := \varphi(\theta) := AB$. At an interpolation point, $h(\phi) = Z$, so the Gauss–Newton identity gives $D^2 f(\theta) = D\varphi(\theta)^\top Dh(\phi)^\top Dh(\phi) D\varphi(\theta)$. Since $dn \geq ab$, standard singular value inequalities yield $\kappa(f)(\theta) \leq \kappa(Dh(\phi))^2 \kappa(D\varphi(\theta))^2$, where $\kappa(D\varphi(\theta))$ is defined similarly as $\kappa(Dh(AB))$ (since $\phi$ also increases dimensions), but with singular values defined on the orthogonal complement of $\ker(D\varphi(\theta))$ (equivalently, by ignoring the zero singular values due to gauge invariance). Proposition 2.3 provides an upper bound on the conditioning of these nonzero singular values, namely $\kappa(D\varphi(\theta))^2 \leq \frac{\sigma_1(A)^2 + \sigma_1(B)^2}{\min(\sigma_r(A)^2, \sigma_r(B)^2)}$, which gives the claimed inequality.

# B. Intrinsic reformulation and structure of the hyperbalanced manifold $\mathcal{H}$

## B.1. Proof of Proposition 3.1

Since $f(A, B) = g(AB)$, the chain rule yields, writing $G_k = \nabla g(X_k)$, $\nabla_A f(A_k, B_k) = G_k B_k^\top$, $\nabla_B f(A_k, B_k) = A_k^\top G_k$. Hence, the pre-projection product is

$$\widetilde{X}_k = \widetilde{A}_k \widetilde{B}_k = (A_k - \tau_k G_k B_k^\top)(B_k - \tau_k A_k^\top G_k)$$
$$= X_k - \tau_k (A_k A_k^\top G_k + G_k B_k^\top B_k) + \tau_k^2 G_k X_k^\top G_k.$$

By construction of the projection $P$, the product is preserved by $P$, hence $X_{k+1} = A_{k+1} B_{k+1} = \widetilde{X}_k$. Since $(A_k, B_k) \in \mathcal{H}$, we take an SVD $X_k = U_k S_k V_k^\top$ and balanced factors $A_k = U_k S_k^{1/2}$ and $B_k = S_k^{1/2} V_k^\top$. Then, $A_k A_k^\top = U_k S_k U_k^\top = (X_k X_k^\top)^{1/2}$, $B_k^\top B_k = V_k S_k V_k^\top = (X_k^\top X_k)^{1/2}$. Substituting into $\widetilde{X}_k$ gives the claimed formula $X_{k+1} = X_k - \tau_k H_{X_k}[G_k] + \tau_k^2 G_k X_k^\top G_k$,

## B.2. Structure of the hyperbalanced manifold

The following proposition further details the structure of the set $\mathcal{H}$.

**Proposition B.1** (Equivalent descriptions of $\mathcal{H}$). *The set $\mathcal{H}$ is a smooth manifold in a neighborhood of full-rank points, with dimension equal to that of the rank-$r$ manifold $\mathcal{N}_r := \{X \in \mathbb{R}^{a \times b} : \text{rank}(X) \leq r\}$. The product mapping $(A, B) \mapsto AB$ is a surjective map from $\mathcal{H}$ onto $\mathcal{N}_r$. If one locally fixes a consistent sign convention for the singular vectors in an SVD decomposition $X = USV^\top$, then the mapping $X \mapsto (US^{1/2}, S^{1/2}V^\top)$ defines a smooth local inverse at points $X$ with non-repeated singular values. Equivalently, $\mathcal{H}$ admits the explicit description*

$$\mathcal{H} = \{(US^{1/2}, S^{1/2}V^\top) : U^\top U = V^\top V = I_r, \ S \in \mathbb{D}_+^r\}. \tag{11}$$

*Proof.* We prove Equation (6). ($\subseteq$) Take $(A, B) \in \mathcal{H}$ with $A^\top A = BB^\top = S \in \mathbb{D}_+^r$. Set $U := AS^{-1/2}$ and $V := B^\top S^{-1/2}$. Then $U^\top U = I_r$, $V^\top V = I_r$, and $A = US^{1/2}$, $B = S^{1/2}V^\top$. ($\supseteq$) Conversely, if $A = US^{1/2}$ and $B = S^{1/2}V^\top$ with $U^\top U = V^\top V = I_r$ and $S \in \mathbb{D}_+^r$, then $A^\top A = S$ and $BB^\top = S$, so $(A, B) \in \mathcal{H}$. $\qquad\square$

*Remark* B.2 (Smoothness of $P$). It is important to note that the definition of $P$ in Equation (7) is not entirely unambiguous: singular vectors are determined only up to sign, and in the presence of repeated singular values they are even invariant under rotations within the degenerate subspace. When analyzing the convergence of optimization schemes over $X = AB$, this ambiguity is harmless, as discussed in Section 3.2. However, to guarantee that $P$ is smooth, one must restrict attention to points $X$ with distinct singular values and adopt a locally consistent sign convention for the singular vectors.

With Remark B.2 in mind, the next proposition shows that $P$ locally enables the definition of a retraction map ((Absil & Malick, 2012)) that preserves the product $AB$.

**Proposition B.3** (Properties of $P$). *Let $P$ be as in Equation (7). Then $P(A, B) \in \mathcal{H}$, and if $(A, B) \in \mathcal{H}$, then $P(A, B) = (A, B)$. Furthermore, $P$ preserves the product $\Pi(A, B) := AB$, i.e., $\Pi(P(A, B)) = \Pi(A, B)$. The map $P$ acts locally as a first–order retraction on $\mathcal{H}$: for any $Z := (A, B) \in \mathcal{H}$ such that $AB$ has distinct singular values, and for any $\Delta \in T_Z \mathcal{H}$ in the tangent plane of $\mathcal{H}$ at $Z$, the map $(Z, \Delta) \mapsto P(Z + \Delta)$ defines a first-order retraction, namely $P(Z + \Delta) = Z + \Delta + o(\Delta)$.*

*Proof.* Fix $(A, B) \in \mathcal{H}$ and assume $Z := AB$ has distinct singular values. By standard perturbation theory for the SVD (with a consistent choice of signs), the reduced SVD $Z \mapsto (U, S, V)$ depends $C^1$-smoothly on $Z$ in a neighborhood of $Z$, hence the map

$$P(A', B') = (U(A'B') S(A'B')^{1/2}, S(A'B')^{1/2} V(A'B')^\top)$$

is $C^1$ in $(A', B')$ near $(A, B)$. Moreover, $P$ fixes $\mathcal{H}$: if $(A', B') \in \mathcal{H}$ then $P(A', B') = (A', B')$.

Let $\Delta \in T_{(A,B)}\mathcal{H}$. By the definition of the tangent space of an embedded submanifold, there exists a $C^1$ curve $\gamma : (-\epsilon, \epsilon) \to \mathcal{H}$ with $\gamma(0) = (A, B)$ and $\dot{\gamma}(0) = \Delta$. Since $P$ fixes $\mathcal{H}$ pointwise, $P(\gamma(t)) = \gamma(t)$ for all $t$ small. Differentiating at $t = 0$ and using the chain rule yields

$$DP(A, B)[\Delta] = \frac{d}{dt}\Big|_{t=0} P(\gamma(t)) = \frac{d}{dt}\Big|_{t=0} \gamma(t) = \Delta.$$

Because $P$ is $C^1$, its first-order expansion at $(A, B)$ gives, for any $\varepsilon \to 0$,

$$P\big((A, B) + \varepsilon\Delta\big) = P(A, B) + \varepsilon\, DP(A, B)[\Delta] + o(\varepsilon) = (A, B) + \varepsilon\Delta + o(\varepsilon).$$

$\square$

## C. Signed-GD conditioning and comparison with RefLoRA

### C.1. Signed-GD conditioning

*Proof of Proposition 2.2.* Fix $\varepsilon > 0$. We split the argument into four steps.

**Step 1: local $\ell^\infty$ smoothness.** Since $f$ is twice continuously differentiable and $\theta_t \to \theta^\star$, the Hessian $\nabla^2 f(\theta)$ is arbitrarily close to $H$ when $\theta$ is sufficiently close to $\theta^\star$. Because the map $M \mapsto \sup_{u \neq 0}\langle Mu, u\rangle/\|u\|_\infty^2$ is continuous on symmetric matrices, there exists a neighborhood $U$ of $\theta^\star$ such that $\sup_{u \neq 0}\langle \nabla^2 f(\eta)u, u\rangle/\|u\|_\infty^2 \leq L_\infty + \varepsilon$ for every $\eta \in U$. Therefore, for all $t$ large enough and all sufficiently small $u$ such that the whole segment $\theta_t + su$ stays in $U$, integrating the Hessian bound along this segment gives the local descent lemma

$$f(\theta_t + u) \leq f(\theta_t) + \langle \nabla f(\theta_t), u\rangle + \frac{L_\infty + \varepsilon}{2}\|u\|_\infty^2$$

for all sufficiently small $u$.

**Step 2: local $\ell^\infty$ Polyak–Lojasiewicz inequality.** Let $\delta_t := \theta_t - \theta^\star$, and decompose it as $\delta_t = \delta_{t,\perp} + \delta_{t,0}$ with $\delta_{t,\perp} \in \ker(H)^\perp$ and $\delta_{t,0} \in \ker(H)$. In our setting, the local minimizer manifold generated by the gauge invariance is tangent to $\ker(H)$ at $\theta^\star$, so the quadratic part of $f$ is governed by the normal component $\delta_{t,\perp}$. More precisely, there exist remainders $r_t$ and $\rho_t$ such that

$$\nabla f(\theta_t) = H\delta_{t,\perp} + r_t, \qquad f(\theta_t) - f(\theta^\star) = \frac{1}{2}\langle H\delta_{t,\perp}, \delta_{t,\perp}\rangle + \rho_t,$$

with $\|r_t\|_1 = o(\|\delta_{t,\perp}\|_2)$ and $\rho_t = o(\|\delta_{t,\perp}\|_2^2)$ as $t \to +\infty$. Consider the scale-invariant ratio $\phi(z) := \|Hz\|_1^2/\langle Hz, z\rangle$ on $\ker(H)^\perp \setminus \{0\}$. Since $\phi$ is continuous on the Euclidean unit sphere of $\ker(H)^\perp$ and this sphere is compact, its minimum is exactly $\mu_\infty$. Therefore, if $\delta_{t,\perp} \neq 0$,

$$\frac{\|\nabla f(\theta_t)\|_1^2}{2(f(\theta_t) - f(\theta^\star))} = \phi(\delta_{t,\perp}) + o(1) \geq \mu_\infty - \varepsilon$$

for all $t$ large enough, while if $\delta_{t,\perp} = 0$ then $\theta_t$ already lies on the local minimizer manifold and the inequality below is trivial. Hence, for all $t$ large enough,

$$\|\nabla f(\theta_t)\|_1^2 \geq 2(\mu_\infty - \varepsilon)\big(f(\theta_t) - f(\theta^\star)\big)$$

**Step 3: one-step contraction.** Now set $u_t := -\gamma\|\nabla f(\theta_t)\|_1 s_t$ with $s_t \in \partial\|\nabla f(\theta_t)\|_1$. By definition of the $\ell^1$ subdifferential, $\langle \nabla f(\theta_t), s_t\rangle = \|\nabla f(\theta_t)\|_1$ and $\|s_t\|_\infty \leq 1$. Since $\nabla f(\theta_t) \to 0$, we have $u_t \to 0$, so for all $t$ large enough the previous local smoothness estimate applies with $u = u_t$. This gives

$$f(\theta_{t+1}) \leq f(\theta_t) - \gamma\|\nabla f(\theta_t)\|_1^2 + \frac{L_\infty + \varepsilon}{2}\gamma^2\|\nabla f(\theta_t)\|_1^2.$$

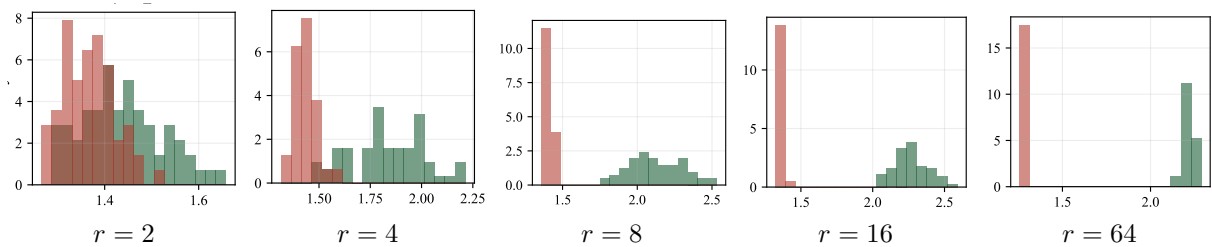

*Figure 6.* Histograms of the scaled-sign-GD condition number $\kappa_\infty = L_\infty/\mu_\infty$, at random balanced (green histograms) and hyperbalanced (brown histograms) minimizers of the matrix factorization loss $f(A, B) = \frac{1}{2}\|Y - AB\|_F^2$. We compare $(A, B) \in \mathcal{B} = \{(A, B) \mid A^\top A = BB^\top\}$ with $(A, B) \in \mathcal{H} = \{(A, B) \in \mathcal{B} \mid A^\top A \in \mathbb{D}_+^r\}$ for $n = m = 200$, ranks $r \in \{2, 4, 8, 16, 64\}$, and independent random rank-$r$ targets $Y$ paired with balanced and hyperbalanced minimizers.

Combining the last two inequalities yields

$$f(\theta_{t+1}) - f(\theta^\star) \leq \left(1 - 2\gamma\left(1 - \frac{\gamma(L_\infty + \varepsilon)}{2}\right)(\mu_\infty - \varepsilon)\right)\left(f(\theta_t) - f(\theta^\star)\right)$$

for all $t$ large enough. Taking the lim sup as $t \to +\infty$, then letting $\varepsilon \downarrow 0$, proves that for every $0 < \gamma < 2/L_\infty$,

$$\limsup_{t \to +\infty} \frac{f(\theta_{t+1}) - f(\theta^\star)}{f(\theta_t) - f(\theta^\star)} \leq 1 - 2\gamma\left(1 - \frac{\gamma L_\infty}{2}\right)\mu_\infty.$$

The right-hand side is minimized at $\gamma = 1/L_\infty$, which gives the factor $1 - \mu_\infty/L_\infty = 1 - 1/\kappa_\infty$.

**Step 4: comparison with Euclidean conditioning.** It remains to compare $\kappa_\infty$ with $\kappa$. For every $v \in \mathbb{R}^D$, one has $\|v\|_2 \leq \|v\|_1 \leq \sqrt{D}\|v\|_2$ and $\|v\|_\infty \leq \|v\|_2 \leq \sqrt{D}\|v\|_\infty$. Hence

$$\mu = \inf_{u \in \ker(H)^\perp \setminus \{0\}} \frac{\|Hu\|_2^2}{\langle Hu, u\rangle} \leq \mu_\infty \leq D \inf_{u \in \ker(H)^\perp \setminus \{0\}} \frac{\|Hu\|_2^2}{\langle Hu, u\rangle} = D\mu,$$

and similarly

$$L = \sup_{u \neq 0} \frac{\langle Hu, u\rangle}{\|u\|_2^2} \leq L_\infty \leq D\sup_{u \neq 0} \frac{\langle Hu, u\rangle}{\|u\|_2^2} = DL.$$

Dividing the two inequalities yields $\kappa/D \leq \kappa_\infty \leq D\kappa$, which completes the proof. $\square$

### C.2. Comparison with RefLoRA

BaLoRA and RefLoRA both enforce balance along the optimization, but they do so on different sets. RefLoRA rebalances the factors so that its iterates satisfy $(A_k, B_k) \in \mathcal{B}$, where $\mathcal{B} := \{(A, B) \mid A^\top A = BB^\top\}$. In contrast, BaLoRA projects onto the smaller hyperbalanced manifold $\mathcal{H} \subset \mathcal{B}$, where $\mathcal{H} := \{(A, B) \in \mathcal{B} \mid A^\top A = BB^\top \in \mathbb{D}_+^r\}$. Thus, RefLoRA is not constrained to stay on $\mathcal{H}$: its iterates remain balanced, but may explore the larger manifold $\mathcal{B}$.

To compare these two geometries through the signed-GD proxy from Proposition 2.2, we consider the matrix factorization loss $f(A, B) = \frac{1}{2}\|Y - AB\|_F^2$ and evaluate the Hessian-based quantity $\kappa_\infty = L_\infty/\mu_\infty$ at random balanced and hyperbalanced global minimizers associated with random rank-$r$ targets $Y$. The resulting histograms are reported in Figure 6. They show that the $\ell^\infty$ geometry relevant to scaled sign-GD can distinguish $\mathcal{B}$ and $\mathcal{H}$.

## D. Additional Empirical Results

We provide in this section some additional figures.

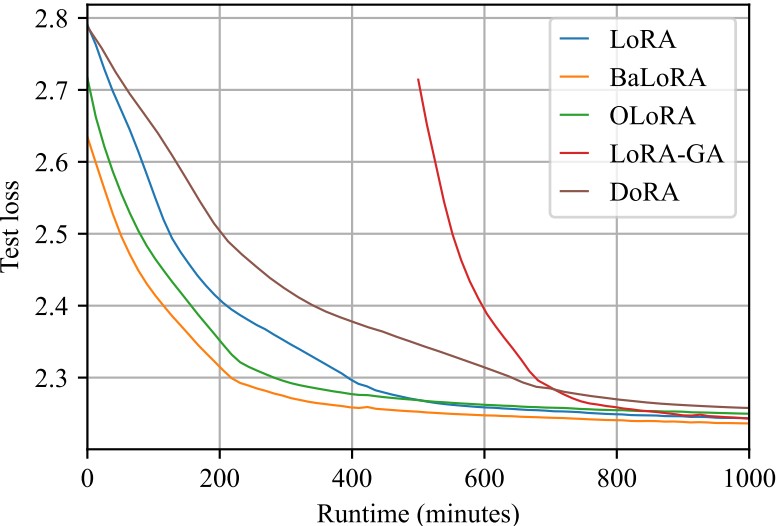

*Figure 7.* Test loss when fine-tuning Llama-3.2-3B on Wikitext as a function of the training time. The initialization time is taken into account; the full gradient estimation in the LoRA-GA init takes $\approx 500$ minutes, which makes it slower than the other methods.

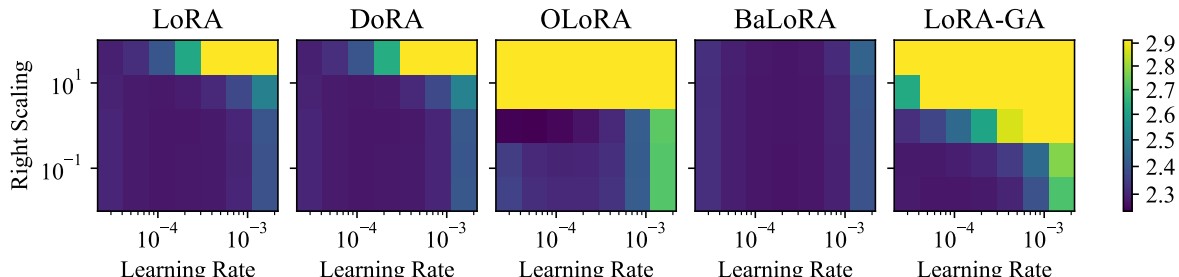

*Figure 8.* Hyperparameter sensitivity analysis of BaLoRA, LoRA and variants for a grid of learning rates and initialization scalings, when fine-tuning Llama-3.2-3B on Wikitext-2-raw-v1. We observe that BaLoRA is significantly more stable to high scalings than the other methods, and more stable to high learning rates than OLoRA and LoRA-GA.

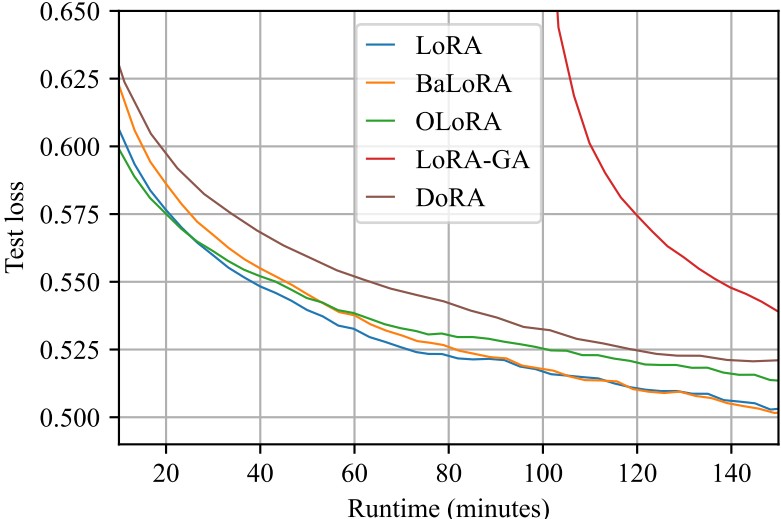

*Figure 9.* Test loss evolution over fine-tuning of Llama-3.2-3B on GSM8K as a function of the training time. The initialization time is taken into account, which explains why LoRA-GA is slower than the other methods.

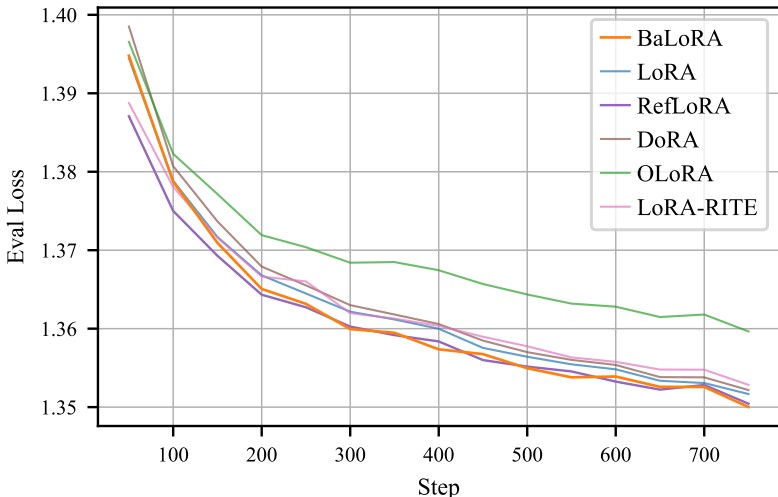

*Figure 10.* Evaluation loss of BaLoRA versus LoRA variants when fine-tuning Qwen-2.5-3B on Alpaca with $r = 8$, selecting optimal per-method and per-dataset hyperparameters based on the MetaMathQA sweep. BaLoRA and RefLoRA, which impose balanced iterations, outperform the other methods.

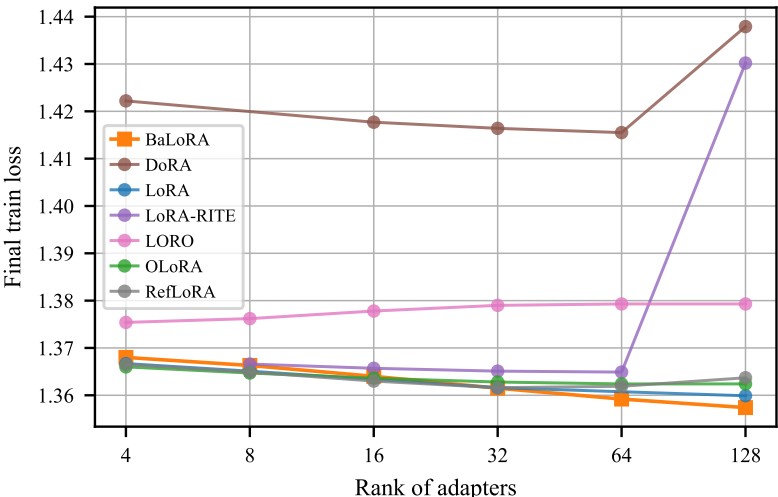

*Figure 11.* Impact of the rank of the adapters on the final train loss when fine-tuning Qwen-2.5-3B on a 1B subset of ArXiv, with the procedure explained in Table 8. LoRA, RefLoRA and OLoRA perform well for small rank values, while BaLoRA outperforms the other methods for larger ranks.

*Table 3.* Results of fine-tuning Llama-3.2-3B on GSM8K. Best loss is in bold, second best loss underlined.

| Method | Epoch 1 | Epoch 2 |
|---|---|---|
| LoRA | 0.498 | 0.492 |
| BaLoRA | 0.506 | 0.493 |
| DoRA | **0.497** | 0.492 |
| OLoRA | 0.510 | 0.503 |
| LoRA-GA | 0.504 | **0.491** |

*Table 4.* Results of fine-tuning Llama-3.2-3B and Qwen-2.5-3B on Wikitext-2-raw-v1. Best loss is in bold, second best loss underlined.

| Method / Model | Llama 3B | Qwen 3B |
|---|---|---|
| LoRA | 2.278 | 2.257 |
| BaLoRA | 2.274 | **2.251** |
| DoRA | 2.278 | 2.257 |
| OLoRA | **2.241** | 2.276 |
| LoRA-GA | 2.276 | 2.264 |

*Table 5.* Results of fine-tuning Qwen-2.5-3B on a 100k-samples subset of MetaMathQA. Best loss is in bold, second best loss underlined.

| Method / Model | Qwen 3B |
|---|---|
| LoRA | **0.142** |
| BaLoRA | 0.144 |
| DoRA | 0.142 |
| OLoRA | 0.144 |
| LoRA-GA | 0.143 |

*Table 6.* Peak GPU memory of LoRA and variants when fine-tuning Llama-3.2-3B on Wikitext-2-raw-v1 with $r = 128$, a batch size of 8, averaged over 10 runs.

| Method | Mean peak GPU memory (GB) | Standard deviation (GB) |
|---|---|---|
| LoRA | 32.63 | 0.03 |
| BaLoRA | 32.63 | 0.03 |
| DoRA | 36.78 | 0.02 |
| OLoRA | 32.63 | 0.03 |
| LoRA-GA | 32.63 | 0.03 |

*Table 7.* Impact of the rank of the adapters on the final train loss when fine-tuning Qwen-2.5-3B on a 1B subset of the DeepMind Mathematics Dataset. For each method, the learning rate and the scaling at initialization are selected from a sweep on MetaMathQA: concretely, we pick the hyperparameters that achieve the smallest evaluation loss when fine-tuning Qwen on MetaMathQA with $r = 8$, which provides a fair procedure for choosing the hyperparameters of each method. Best loss is in bold, second best loss underlined. BaLoRA is the best method for almost all ranks, and has a clear edge for larger ranks.

| Method | $r = 2$ | $r = 4$ | $r = 8$ | $r = 16$ | $r = 32$ | $r = 64$ | $r = 128$ |
|---|---|---|---|---|---|---|---|
| LoRA | 1.050 | 1.041 | 1.035 | 1.032 | 1.031 | 1.030 | 1.030 |
| DoRA | 1.050 | 1.041 | 1.035 | 1.032 | 1.031 | 1.030 | 1.030 |
| LoRA-RITE | 1.059 | 1.051 | 1.047 | 1.045 | 1.046 | 1.052 | 1.069 |
| LoRO | 1.251 | 1.253 | 1.258 | 1.266 | 1.271 | 1.272 | 1.267 |
| OLoRA | 1.051 | 1.044 | 1.039 | 1.037 | 1.036 | 1.036 | 1.036 |
| RefLoRA | **1.046** | 1.035 | 1.027 | 1.023 | 1.024 | 1.027 | 1.032 |
| BaLoRA | 1.046 | **1.035** | **1.026** | **1.020** | **1.017** | **1.015** | **1.014** |

*Table 8.* Impact of the rank of the adapters on the final train loss when fine-tuning Qwen-2.5-3B on a 1B subset of ArXiv. For each method, the learning rate and the scaling at initialization are selected from a sweep on MetaMathQA: concretely, we pick the hyperparameters that achieve the smallest evaluation loss when fine-tuning Qwen on MetaMathQA, which provides a fair procedure for choosing the hyperparameters of each method. Best loss is in bold, second best loss underlined. We observe that LoRA, DoRA and LoRA-RITE are less robust to transferring hyperparameters across a wide range of ranks: they diverge for some rank values. Moreover, BaLoRA has a clear edge for larger ranks.

| Method | $r = 2$ | $r = 4$ | $r = 8$ | $r = 16$ | $r = 32$ | $r = 64$ | $r = 128$ |
|---|---|---|---|---|---|---|---|
| LoRA | 1.368 | 1.367 | 1.365 | 1.399 | 1.362 | NaN | 1.360 |
| DoRA | 1.423 | 1.422 | NaN | 1.418 | 1.416 | 1.416 | 1.438 |
| LoRA-RITE | 1.369 | NaN | 1.367 | 1.366 | 1.365 | 1.365 | 1.430 |
| LoRO | 1.378 | 1.375 | 1.376 | 1.379 | 1.379 | 1.379 | 1.379 |
| OLoRA | **1.367** | **1.366** | **1.365** | 1.364 | 1.363 | 1.362 | 1.362 |
| RefLoRA | 1.368 | 1.367 | 1.365 | **1.363** | 1.362 | 1.362 | 1.364 |
| BaLoRA | 1.369 | 1.368 | 1.366 | 1.364 | **1.362** | **1.360** | **1.357** |

