# OpenReview forum: "Balanced LoRA: Removing Parameter Invariance to Accelerate Convergence"
_ICML.cc/2026/Conference — ICML 2026 regular_

### Official Review · Reviewer_kCtx · 2026-02-16

**Soundness:** 4
**Presentation:** 3
**Significance:** 3
**Originality:** 2
**Overall Recommendation:** 5
**Confidence:** 3

**Summary:**

In this paper, the authors analyze that in Low-Rank Adaptation (LoRA), multiple pairs of low rank factors have different condition numbers. Therefore, it may converge to different loss minimizers and impact the convergence rate of LoRA. To address this issue, the authors propose Balanced Low-Rank Adaptation (BaLoRA). BaLoRA can project iterates in a balanced manifold. Therefore, BaLoRA can converge faster than LoRA.

**Compliance With Llm Reviewing Policy:**

Affirmed.

**Final Justification:**

My concerns have been fully addressed.

**Key Questions For Authors:**

Is it possible to use other methods to control the condition number, for example sketch-and-precondition?

**Limitations:**

Yes

**Strengths And Weaknesses:**

The strengths of this paper are summarized as follows:

$\bullet$ This paper analyzes an interesting question that identifies a source of optimization inefficiency in LoRA. This paper is well-written and easy to follow.

$\bullet$ The theoretical analysis is convincing and sound. It shows that minimizers lie on a continuous manifold, and the conditioning of the loss changes along this manifold. This paper shows that the balanced minimizer $(A, B)$ satisfies $A^\top A = BB^\top$. Therefore, a good conditioning may lead to better convergence.

$\bullet$ In the experiment, the authors evaluate multiple large language models, GPT-2, Llama-3.2-3B, and Qwen-2.5-3B across various tasks. This paper shows that BaLoRA converges faster after an initial slower phase.


The weaknesses of this paper are summarized as follows:

$\bullet$ I believe the main contribution of this paper is on theoretical and structural sides. The authors acknowledge that

> this balance condition has been studied in other contexts, such as linear networks (Nguegnang et al., 2024), matrix factorization (Ye & Du, 2021; Ghosh et al., 2025), and conservation laws in neural networks (Marcotte et al., 2023).

While the application to LoRA is interesting, the paper would benefit from a more detailed technical comparison clarifying what aspects of the conditioning analysis are fundamentally new versus adaptations of existing balancing theory.

$\bullet$ This paper does present a comparison with other LoRA variants, but it does not show its advantage over other parameter-efficient fine-tuning methods (PEFTs). Numerous PEFTs have been developed over the years. I agree that this paper provides a valuable advancement over LoRA, but it would be better to compare the results with other PEFT methods, such as Galore.

---

> ### Author Rebuttal · Authors · 2026-03-30
>
> We thank the reviewer for acknowledging that our paper is "well-written and easy to follow", "analyzes an interesting question", with a "convincing and sound" theoretical analysis.
>
> > **The authors evaluate multiple large language models across various tasks**
>
> We are glad to share that we have performed **two additional main experiments** on Qwen2.5-3B:
> * Dataset sweep: we evaluate LoRA, BaLoRA, DoRA, OLoRA, LoRA-RITE, RefLoRA, and LORO across Alpaca, Code-Feedback, OpenHermes, OpenOrca, and WizardLM. In that experiment, most methods achieve very similar eval losses; **Balora is consistently among the top performing methods**.
> * Rank sweep: we vary the LoRA rank from 2 to 128 on two datasets, Arxiv and DM-Mathematics. **Balora significantly outperforms the other methods for the larger ranks** (64, 128), which illustrates the potential of our method for fine-tuning with large rank and lots of data.
>
> The results are here: https://anonymous.4open.science/r/ICML_2026_rebuttal-D412/icml2026_rebuttal.pdf
>
> > **the paper would benefit from a more detailed technical comparison clarifying what aspects of the conditioning analysis are fundamentally new versus adaptations of existing balancing theory.**
>
> We appreciate this comment, as it helps clarify an important conceptual point. What is inherited from prior work is the notion of balancedness itself, and the fact that factorized parameterizations of a low-rank update $\Delta W = AB$ admit nontrivial reparameterization invariances. What is new here is to **connect balancedness to the conditioning of the loss**. The objective has the form $f(W_0 + \Delta W)$, where only the low-rank update $\Delta W = AB$ is trained. Hence, all pairs $(A,B)$ in the orbit {$(AR, R^{-1}B), R \in \mathbb{R}^{r \times r} \text{ invertible}$} represent the same $\Delta W$, but **they do not induce the same conditioning of the optimization problem in factor space**.
>
> To our knowledge, **this connection between balancedness and conditioning has not been studied for LoRA**. The closest related work is Ghosh et al. (2025), which studies deep matrix factorization under a stronger structural condition (balancing singular values across all layers, see their Section 3.1.2), and does not show that balanced factorizations are optimal in terms of conditioning. In contrast, we show that among all equivalent LoRA factorizations representing the same update $\Delta W$, the balanced ones (where $A^\top A = B B^\top$) provably achieve the best conditioning. This is precisely what makes balancing relevant in LoRA: it is not merely an invariant-related structural property, but a principled way to select the best-conditioned parameterization of a fixed adapted matrix.
>
> > **I agree that this paper provides a valuable advancement over LoRA, but it would be better to compare the results with other PEFT methods, such as Galore.**
>
> We agree that broader PEFT context is useful. However, methods such as GaLore are not directly comparable to the adaptation budget of LoRA-style methods.
>
> Specifically, unlike LoRA, GaLore does not produce an adapter with **far fewer parameters** than the base model; instead, it utilizes low-rank gradient projections to update a much larger parameter set, often the full weights themselves. BaLoRA, by contrast, is designed as a strict drop-in replacement for LoRA: it preserves the frozen-backbone PEFT setup and maintains the exact same trainable parameter count of $r(a+b)$. While GaLore is a valuable reference for memory-efficient training, it operates in a different regime and is not an "apples-to-apples" replacement for a rank-$r$ adapter method in terms of deployment footprint. We have therefore focused on LoRA-type baselines that share the same structural budget and modular deployment setting. Still, the reviewer is right that readers would benefit from a clearer discussion of where BaLoRA sits within the wider PEFT landscape. We will add such a discussion in the revised version.
>
> > **Is it possible to use other methods to control the condition number, for example sketch-and-precondition?**
>
> Yes, thank you for the great question. In fact, **BaLoRA can already be interpreted as a geometry-aware preconditioned method** on the low-rank matrix space.
>
> The intrinsic formulation in the paper shows that, for $X=AB$, BaLoRA-GD updates $X$ through the operator $H_X[W]=(XX^\top)^{1/2}W + W(X^\top X)^{1/2}$, which is precisely a structured, state-dependent preconditioning of the Euclidean gradient. In this sense, balancing is not only a constraint on factor norms; it induces a specific geometry on the rank-$r$ manifold. Sketching, approximate preconditioning, or hybrid preconditioned schemes are therefore very plausible extensions and may be especially interesting at higher ranks.
> We will add this perspective in the discussion.
>
> We thank you again for your comments, which help us improve the paper, and hope that our answers clarify our contributions.

---

> > ### Author Rebuttal · Reviewer_kCtx · 2026-04-03
> >
> > Thank you very much for your clarification. My concerns have been fully addressed.

---

### Official Review · Reviewer_MeGz · 2026-02-23

**Soundness:** 2
**Presentation:** 3
**Significance:** 2
**Originality:** 2
**Overall Recommendation:** 3
**Confidence:** 4

**Summary:**

The paper argues that enforcing the condition $A^\top A = B^\top B$ the low-rank update to optimize the loss landscape. They provide theoretical proofs showing that this balanced state minimizes the condition number in single-layer LoRA and some of its variants. The improved condition number can lead to improved optimization convergence. Numerical examples are also provided.

**Compliance With Llm Reviewing Policy:**

Affirmed.

**Final Justification:**

My concerns on experiments are addressed.

**Key Questions For Authors:**

Q1. Proposition 2.6 is established only for minimizers. However, once the optimization has reached a minimizer, the process should in principle terminate. In that case, it is unclear how analyzing the Hessian at a minimizer provides insight into the optimization dynamics. Could the authors clarify how this result contributes to understanding or improving optimization?


Q2. Could the authors provide additional comparisons of memory consumption for BA-LoRA relative to other methods?

**Strengths And Weaknesses:**

**Strength**

(+) Theoretical analyses on relevant examples demonstrate that balancing A and B improves the condition number. These results offer a theoretical justification for adopting balanced A/B parameterizations in LoRA fine-tuning.

**Weakness**

(-) The primary weakness of this work lies in its novelty. Prior studies, such as [R1, R2], have already introduced balanced A and B parameterizations, albeit motivated by different arguments. However, the manuscript does not provide a sufficiently in-depth discussion of these related works, making it unclear how the present contribution meaningfully advances beyond the existing literature.

(-) The reported numerical gains seem limited. In addition, Fig. 4 seems to suggest that BaLoRA uses a different initialization scheme from LoRA, and its initial loss is even lower than that of LoRA-GA, which is explicitly designed for better initialization. This discrepancy makes the comparison appear unfair.

[R1] https://arxiv.org/abs/2505.18877

[R2] https://arxiv.org/abs/2407.18242

---

> ### Author Rebuttal · Authors · 2026-03-30
>
> We thank the reviewer for their feedback and positive appreciation of our "theoretical analyses on relevant examples".
>
> > **it is unclear how analyzing the Hessian at a minimizer provides insight into the optimization dynamics.**
>
> Our argument follows a standard approach in optimization: *to characterize the asymptotic convergence rate, one studies the local behavior of the objective near a minimizer* (see e.g., Sec. 9 of "Convex Optimization" by Boyd and Vandenberghe).
>
> More precisely, one approximates the loss locally by the quadratic function given by its second-order Taylor's expansion around the minimizer of the trajectory. Hence, once the iterates are sufficiently close to their limit $(A^\star, B^\star)$, the convergence rate in this regime (referred to in the paper as the *asymptotic convergence rate*) is governed by the conditioning of the Hessian of the loss at $(A^\star,B^\star)$ (Lemma 2.1). Thank you for this question, we will clarify this point in the revised version.
>
> > **Prior studies, such as [R1, R2], have already introduced balanced A and B parameterizations**
>
> Thanks a lot for the relevant references. Our contribution is **distinct from both works**, both algorithmically and theoretically. We will revise the related work discussion to make these distinctions explicit.
>
> * LoRA-Pro [R2] and BaLoRA are based on different principles. LoRA-Pro modifies the gradients of $A$ and $B$ so that the induced update on $\Delta W=AB$ better approximates the gradient update on the full weight $W = W_0 + \Delta W$. In contrast, BaLoRA leaves the gradients unchanged and instead rebalances $A$ and $B$ during training. Thus, the two methods significantly differ in both their update rules and their motivation: LoRA-Pro is based on gradient correction, while BaLoRA is driven by balancing the low-rank factors.
> * RefLoRA [R1] is closer to BaLoRA, since both methods maintain balanced factors $(A_t, B_t)$ through optimization. However, the balancing is obtained differently. RefLoRA uses $$ \tilde P (A,B)= (AS(A,B)^{1/2}, S(A,B)^{-1/2}B)$$ with $S(A,B) = (A^\top A)^{-1/2}[(A^\top A)^{1/2} (B B^\top)(A^\top A)^{1/2}]^{1/2}(A^\top A)^{-1/2},$ whereas BaLoRA applies $$P(A,B) = (U\Sigma^{1/2}, \Sigma^{1/2}V^\top)$$ with $U\Sigma V^\top$ the singular value decomposition of $AB$. In general, $\tilde P(A,B) \neq P(A,B)$, thus the two methods do not produce the same balanced iterates. In particular, they induce different Adam updates and **different optimization dynamics**.
>
> Following your suggestion, we have added RefLoRA to our experiments (see below): **BaLoRA significantly outperforms RefLoRA for larger adapter ranks**.
>
> Finally, our main contribution is theoretical: **neither LoRA-Pro nor RefLoRA provides an analysis of the conditioning of balanced minimizers** and its consequences for asymptotic convergence.
>
> > **The reported numerical gains seem limited**
>
> We have **added two main experiments**:
> - Dataset sweep on Qwen2.5-3B. We evaluate LoRA, BaLoRA, DoRA, OLoRA, LoRA-RITE, RefLoRA, and LORO across Alpaca, Code-Feedback, OpenHermes, OpenOrca, and
>   WizardLM. For each method, the learning rate and initialization scaling are selected from a 10×10 grid sweep on MetaMathQA, then applied to all datasets. In that experiment, most methods achieve very similar eval losses; **Balora is consistently among the top performing methods**.
> - Rank sweep on Qwen2.5-3B. We vary the LoRA rank in {2, 4, 8, 16, 32, 64, 128} with per-method optimal hyperparameters fixed as above. We run this on two datasets, Arxiv and DM-Mathematics, training for 1B tokens. In that experiment, **Balora achieves significantly better results than the other methods for the larger ranks** (64, 128). This illustrates the potential of our method for fine-tuning with large rank and lots of data.
>
> The results are available here: https://anonymous.4open.science/r/ICML_2026_rebuttal-D412/icml2026_rebuttal.pdf
>
> > **BaLoRA uses a different initialization scheme from LoRA**
>
> We believe there is a misunderstanding: BaLoRA uses **the same initialization scheme** as LoRA (see line 286, right column). The initial loss in Fig. 4 corresponds to the loss **after 10 training steps**, because the x axis starts at the first evaluation, which occurs after 10 training steps. This explains the differences across methods: BaLoRA decreases the loss faster than LoRA over the first 10 iterations, while starting from the *same loss value* with the *same random seed*. The same comment applies for LoRA-GA. We will clarify this in the text.
>
> > **Could the authors provide additional comparisons of memory consumption for BA-LoRA relative to other methods?**
>
> Thank you for this suggestion. We have **compared the Peak GPU memory for several methods**, averaged over 10 runs, when fine-tuning Llama on Wikitext. The results (Table 4 in the file above) show that **BaLoRA and LoRA have the same memory consumption.**
>
> We thank you again for your comments, which help us improve the paper.

---

> > ### Author Rebuttal · Reviewer_MeGz · 2026-04-02
> >
> > Thanks for the detailed responses, especially the additional numerical experiments. i have adjusted the score accordingly.
> >
> > Follow up questions: Since both BaLoRA and RefLoRA [R1] make use of balancing, and other balanced variants may also be possible, it would be helpful to understand whether there are practical considerations that would favor one design over another.

---

> > > ### Author Response · Authors · 2026-04-07
> > >
> > > > Since both BaLoRA and RefLoRA [R1] make use of balancing, and other balanced variants may also be possible, it would be helpful to understand whether there are practical considerations that would favor one design over another.
> > >
> > > Thank you for this follow-up question, which gives us the opportunity to clarify the  distinction between BaLoRA and RefLoRA. We identify three main differences between the two methods:
> > >
> > > - **Practical performance.** From an experimental perspective, BaLoRA and RefLoRA perform similarly for small rank values, while BaLoRA significantly outperforms RefLoRA in the high rank / long sequence length regime, and **the gap widens as the factorization rank increases** (Figures 2 and 3 in https://anonymous.4open.science/r/ICML_2026_rebuttal-D412/icml2026_rebuttal.pdf). This indicates that the two methods, while both based on balancing, are **not interchangeable: BaLoRA shows a clear practical advantage in the high rank / long sequence length regime**.
> > >
> > > - **Theoretical distinction.** The key difference is that RefLoRA projects the iterates  onto the balanced manifold $\mathcal{B}$, whereas BaLoRA projects them onto the strictly  smaller hyperbalanced manifold $\mathcal{H} =${$(U\Sigma^{1/2}, \Sigma^{1/2}V^\top) : U^\top U = V^\top V = I_r$} $\subsetneq \mathcal{B}$. Our theory shows that, in the matrix factorization setting, the Euclidean condition number $\kappa(H) =
> > >   \frac{\lambda^{\max}(H)}{\lambda^{\min}(H)}$ is optimal on both $\mathcal{H}$ and $\mathcal{B}$. However, as discussed in our response to Reviewers dK5e and ULNQ, when  optimizing with Adam, the Euclidean condition number may not fully capture the relevant  optimization geometry. Instead, the $\ell^\infty$ condition number $\kappa^\infty(H) =
> > >   \frac{\lambda^\max_\infty(H)}{\lambda^\min_\infty(H)}$ with $\lambda_\infty^{\max}(H) =
> > >   \max_{x \neq 0} \frac{\lVert Hx \rVert_1}{\lVert x \rVert_\infty}$ (replace max by min for $\lambda_\infty^{\min}$) may provide a more informative perspective for Adam-type optimizers.
> > > We currently do not have a theoretical proof that the $\ell^\infty$ conditioning is systematically more favorable on $\mathcal{H}$ than on $\mathcal{B}$ for large ranks.  Nevertheless, as detailed hereafter, our additional experiments on random matrix factorization consistently suggest that this is indeed the case: **the $\ell^\infty$ condition number is typically smaller on $\mathcal{H}$ than on $\mathcal{B}$, and the gap increases with the factorization rank**.
> > >
> > > - **New numerical experiment on Adam-aware conditioning.** Following your suggestion, we have added a new experiment to illustrate the previous point: see Figure 5 in https://anonymous.4open.science/r/ICML_2026_rebuttal-D412/icml2026_rebuttal.pdf. On random matrix factorization instances, the $\ell^\infty$ condition number is consistently smaller on $\mathcal{H}$ than on $\mathcal{B}$, and the gap increases with the factorization rank. Based on these observations, we believe that **improved $\ell^\infty$-type conditioning is the reason BaLoRA achieves stronger empirical performance with Adam**.
> > >
> > > Thank you again for this question, which sharpens our analysis and will be  incorporated in the final version of the manuscript. Since this is our last opportunity to respond, we hope that the added explanation, along with the new experiments, clarify the practical and theoretical reasons favoring BaLoRA over RefLoRA.

---

### Official Review · Reviewer_ULNQ · 2026-03-03

**Soundness:** 3
**Presentation:** 3
**Significance:** 2
**Originality:** 2
**Overall Recommendation:** 4
**Confidence:** 3

**Summary:**

The paper proposes BaLoRA, which aims to acclerate the convergence of LoRA by addressing its inherent overparameterization. The authors provide a theoretical analysis of the asymptotic convergence rate, demonstrating that balanced minimizers ($A^\top A = BB^\top$) achieve the optimal condition number for the loss Hessian, leading to faster training. Experiments show BaLoRA performs better than LoRA when fine-tuning Llama and GPT-2,  but perform not good when fine-tuning QWen 2.5.

**Compliance With Llm Reviewing Policy:**

Affirmed.

**Final Justification:**

The authors give an explanation to the issue of convergence analysis and inconsistency between Adam's per-element moment estimates and balancing projections. This has resolved my concerns.

**Key Questions For Authors:**

Could you please explain the detailed algorithm of BaLoRA (with Adam), such as how the first-order and second-order momentum are integrated?

**Limitations:**

The paper did not discuss the limitations.

**Strengths And Weaknesses:**

### Strengths

+ Novel theoretical grounding: The paper has a theoretical analysis of how balance (A, B) can minimize the upper bound of the condition number of the Hessian matrix of loss function.

+ Linear and non-linear analysis: The thoretical analysis contains linear and non-linear case.
+ The experiments show BaLoRA performs better than LoRA when fine-tuning Llama and GPT-2.

### Weakness

+ Incomplete of baselines: I find that two critical related works, named LoRA-Done-RITE [1] and RefLoRA [2]. While the author cite LoRA Done RITE, but the authors did not provide a experimental comparison, which is necessary to distinguish the "balanced manifold" approach from "invariant transformations." Besides, Table 1 and 2, Figure 8 only included one or two baselines.
+ Gap between theoretical and practice: The theoretical analysis is based on gradient descent, but in the modern LLM training, AdamW is the first choice.
+ Convergence visualization: The paper relies on the final training/test loss and "loss - training time". The comparison of convergence rate ''loss-training step" would better support the claim of "acclerate convergence".
+ Rank sensitivity: The results primarily reported for rank=8. Experiments for other ranks such as 4, 16 or 64 were not concluded.

[1] Yen, J.N., Si, S., Meng, Z., Yu, F., Duvvuri, S.S., Dhillon, I.S., Hsieh, C.J. and Kumar, S., LoRA Done RITE: Robust Invariant Transformation Equilibration for LoRA Optimization. In *The Thirteenth International Conference on Learning Representations*.

[2] Zhang, Y., Li, B. and Giannakis, G.B., RefLoRA: Refactored Low-Rank Adaptation for Efficient Fine-Tuning of Large Models. In *The Thirty-ninth Annual Conference on Neural Information Processing Systems*.

---

> ### Author Rebuttal · Authors · 2026-03-30
>
> We thank the reviewer for their feedback and for highlighting the "novel theoretical grounding" of our paper, with a "linear and non-linear analysis".
> > **Incomplete baselines**
>
> > **Rank sensitivity**
>
> Thanks a lot for these comments and relevant references. We have **added two main experiments**:
> - Dataset sweep on Qwen2.5-3B. We evaluate LoRA, BaLoRA, DoRA, OLoRA, LoRA-RITE, RefLoRA, and LORO across Alpaca, Code-Feedback, OpenHermes, OpenOrca, and
>   WizardLM. For each method, the learning rate and initialization scaling are selected from a 10×10 grid sweep on MetaMathQA, then applied to all datasets. In that experiment, most methods achieve very similar eval losses; **Balora is consistently among the top performing methods**.
> - Rank sweep on Qwen2.5-3B. We vary the LoRA rank in {2, 4, 8, 16, 32, 64, 128} with per-method optimal hyperparameters fixed as above. We run this on two datasets, Arxiv and DM-Mathematics, training for 1B tokens. In that experiment, **Balora achieves significantly better results than the other methods for the larger ranks** (64, 128). This illustrates the potential of our method for fine-tuning with large rank and lots of data.
>
> The results are at https://anonymous.4open.science/r/ICML_2026_rebuttal-D412/icml2026_rebuttal.pdf
>
> > **Gap between theoretical and practice: The theoretical analysis is based on gradient descent, but in the modern LLM training, AdamW is the first choice**
>
> Theoretically analyzing the convergence of Adam is still out of reach in the modern literature: Adam lacks convergence guarantees even for convex quadratic objectives (Reddi, Kale, Kumar 2018). However, in the vanishing-momentum limit $x_{t+1} = x_t -\tau\, \mathrm{sign}(\nabla f(x_t))$, which is the prominent theoretical model for Adam, a simple argument shows that a small conditioning (which is achieved for balanced minimizers independently of the optimizer, according to our theoretical results) allows to reach a smaller loss value. Indeed, if $x^\star$ is a nondegenerate minimizer and $H = D^2f(x^\star)$, writing $f(x)\approx \frac12 (x - x^\star)^\top H (x-x^\star) + f(x^\star)$ in the asymptotic regime gives, for $s_t:= \mathrm{sign}(H(x_t-x^\star))$:
> $$f(x_{t+1}) - f(x_t) \approx \tau \lVert H(x_t - x^\star)\rVert_1 +\frac{\tau^2}{2}s_t^\top H s_t \le -\tau\sqrt{2\lambda_\infty^{\mathrm{min}}(H) f(x_t)} + \frac{\tau^2}{2}\lambda_\infty^\mathrm{max}(H),$$
> where $\lambda_\infty^{\mathrm{min}}(H):=\min_{x\neq 0} \frac{\lVert Hx\rVert_1}{\lVert x\rVert_\infty}$ (replace min by max for $\lambda_\infty^\mathrm{max}$). Accordingly, the relevant notion is not asymptotic contraction to the minimizer, as in ordinary GD, but the terminal oscillation floor: the error level at which the negative drift and the positive discretization term become comparable, so that the iterates stop decreasing monotonically and begin to oscillate around $x^\star$. Monotone decrease is ensured as long as $f(x_t)>\tau^2 d\lambda_\infty^\mathrm{max}(H)\kappa(H)$  in dimension $d$, where $\kappa(H)$ is the spectral conditioning of $H$. Hence, a smaller $\kappa(H)$ leads to a smaller minimum error floor.
>
> Many thanks for your question: this remark, that we will add in the revised version, **significantly broadens the impact of our theoretical results.**
>
> > **The comparison of convergence rate ''loss-training step" would better support the claim of "acclerate convergence".**
>
> Following this suggestion, **we provide a loss-training step version of Fig. 4** at the above link (Fig. 4). With this convention, BaLoRA is second best. We have also taken this convention for Fig 1 (fine-tuning Qwen on Alpaca) to better support the claim of accelerated convergence. We will add all these figures to the revised version.
>
> > **detailed algorithm of BaLoRA (with Adam)**
>
> BaLoRA with AdamW is *standard AdamW* on $(A,B)$ *followed by the balancing projection* $P$ after each parameter update.
>
> We maintain AdamW moment buffers $(m_A,v_A,m_B,v_B)$. At iteration $t$, starting from the current balanced pair $(A_t,B_t)$, we compute the gradients $\nabla_A f(A_t,B_t)$ and $\nabla_B f(A_t,B_t)$, update the first and second moments, and apply the usual AdamW step to obtain an unconstrained pair $(\tilde A_{t+1},\tilde B_{t+1})$. We then set $(A_{t+1},B_{t+1}) = P(\tilde A_{t+1},\tilde B_{t+1}).$ This keeps the parameterization and optimizer-state size identical to LoRA; the only additional operation is the projection. The moment buffers are maintained in the factor coordinates as in projected adaptive optimization, which is simple to implement and works robustly in our experiments.
> We **will add explicit pseudocode for BaLoRA with AdamW** for clarity.
>
> We thank you for your questions which help us improve the paper, and hope that our answers clarify our contributions.

---

> > ### Author Rebuttal · Reviewer_ULNQ · 2026-04-03
> >
> > Thanks for the detailed response. I still have some follow-up questions for the algorithm BaLoRA (Adam).
> >
> > Thank you for the signGD analysis. However, the convergence of Adam has been established under non-convex conditions in [1] and [2].
> >
> > Additionally, in BaLoRA (Adam), the balancing projection maps (A, B) to a different factorization (A', B') while preserving AB. However, Adam's per-element moment estimates are tied to a specific factorization and do not transform accordingly, causing gradient statistics from incompatible factorization pairs to be accumulated across steps. Could the authors comment on this potential issue?
> >
> > [1] Défossez, Alexandre, Leon Bottou, Francis Bach, and Nicolas Usunier. "A Simple Convergence Proof of Adam and Adagrad." Transactions on Machine Learning Research.
> >
> > [2] Zhang, Yushun, Congliang Chen, Naichen Shi, Ruoyu Sun, and Zhi-Quan Luo. "Adam can converge without any modification on update rules." Advances in neural information processing systems 35 (2022): 28386-28399.

---

> > > ### Author Response · Authors · 2026-04-07
> > >
> > > Thank you for these follow-up questions.
> > >
> > > **On Adam convergence results [1, 2].** Thanks for these references. Papers [1,2] indeed provide important results on the convergence guarantees of Adam, and we will include them in the final version of the paper. This being said, our claim is *not* that Adam fails to converge in general: the results of [1, 2] indeed establish that in the $L$-smooth setting, Adam converges in the non-convex sense (i.e., the averaged gradient norm decreases at rate $O(\ln(N)/\sqrt{N})$). However, these guarantees are of a fundamentally different nature from our conditioning-dependent analysis. Specifically, the bounds in [1, 2] characterize *global* convergence to approximate stationary points, and do *not* describe *how the local geometry (conditioning) of the loss landscape around a minimizer affects the convergence speed*. Our analysis operates at a complementary level: we study the asymptotic regime where iterates are close to a minimizer, and show that the conditioning of the Hessian at that minimizer governs the local convergence behavior. We would like to acknowledge that, while this analysis is classical for gradient descent (Boyd & Vandenberghe, Sec. 9; see our first answer to reviewer MeGz), its applicability to Adam is less straightforward due to the method's adaptive scaling. As discussed in our second response to MeGz, a related analysis can be carried out for sign-GD using an $\ell^\infty$ condition number; this method can be interpreted as a simplified version of Adam in the vanishing momentum limit [3]. Our additional numerical results indicate that **BaLoRA also improves this $\ell^\infty$ condition number**.
> > >
> > > We will add this clarification and cite [1, 2] in the final version.
> > >
> > > **On the interaction between balancing projection and Adam's moment estimates.** This is an insightful observation. Indeed, after the balancing projection $(A, B) \mapsto (A', B')$, the Adam moment buffers were accumulated under the previous factorization and do not exactly correspond to the new one. However, we emphasize that this mismatch does not appear to cause practical difficulties:
> > >
> > > - **Empirical validation.** Our experiments consistently show that BaLoRA with AdamW converges *faster* and to better solutions than LoRA with AdamW across multiple models and tasks. This suggests that the moment mismatch *does not cause practical issues*. We also note that replacing the projection with gradient preconditioning (which would eliminate the mismatch entirely but only ensure approximate balancing) is a possible variant, but we found it unnecessary in our experiments.
> > >
> > > - **This mismatch is common to all projected adaptive methods.** Many widely used methods in the litterature face the exact same mismatch because they modify the iterates outside the optimizer step (such as weight projection [4], weight clipping [5]). Despite this mismatch, they are routinely successfully used by practitioners.
> > >
> > > We agree that a more detailed theoretical analysis of the interplay between balancing projections and adaptive moment estimation is an interesting direction for future work. We will add a discussion of these point in the revised version.
> > >
> > > Since this is our last opportunity to respond, we hope our answers clarify these points. Thank you very much for the thoughtful questions, which have helped us improve the paper.
> > >
> > > **References:**
> > >
> > > [3] T. Pethick, W. Xie, K. Antonakopoulos, Z. Zhu, A. Silveti-Falls, and V. Cevher, Training Deep Learning Models with Norm-Constrained LMOs, ICML 2025.
> > >
> > > [4] T. Miyato, T. Kataoka, M. Koyama, Y. Yoshida, Spectral normalization for generative adversarial networks, ICLR 2018
> > >
> > > [5] Gulrajani, I., Ahmed, F., Arjovsky, M., Dumoulin, V., & Courville, A. C. (2017). Improved training of wasserstein gans. Advances in neural information processing systems

---

### Official Review · Reviewer_dK5e · 2026-03-12

**Soundness:** 4
**Presentation:** 4
**Significance:** 4
**Originality:** 4
**Overall Recommendation:** 5
**Confidence:** 4

**Summary:**

In this paper, the authors analyzed the impact of LoRA's over-parameterization from the perspective of loss conditioning and provided a geometric interpretation. The authors revealed the condition number problem underlying LoRA's over-parameterization and tried to address it through simple projection. Then the authors proposed BALoRA, a lightweight LoRA variant that projects low rank matrices (A, B) onto hyperbalanced manifold after each optimization step. This project preserves the product of AB and steers training toward well-conditioned minimizers.

**Compliance With Llm Reviewing Policy:**

Affirmed.

**Final Justification:**

In the rebuttal, the authors have addressed my concerns, so I keep my positive scores.

**Key Questions For Authors:**

Please see Weaknesses.

**Limitations:**

No. the authors should analyze the gap between using GD and AdamW in BaLoRA, and BaLoRA can only advance marginally compared with LoRA.

**Strengths And Weaknesses:**

Strengths:
1. The paper is well-organized and easy to follow;

2. The mathematical analysis is rigorous;

3. The authors made several claims in the paper:(1) Theoretically and empirically, LoRA factors exhibit significantly different condition numbers; (2) There exists a disadvantage in LoRA: converging to different loss minimizers directly impacts the convergence rate of LoRA; (3) BaLoRA projects iterates onto a balanced manifold, which improves the conditioning of the loss landscape and preserves the adapted matrix; (4) The projection step is computationally lightweight and integrates seamlessly into exisitng PEFT  methods; (5) BaLoRA converges faster than LoRA. I found that these claims are all supported by theoretical analysis or experiments. This work is solid.

4. The computational complexity of BALoRA is O((a+b)r^2), which is negiligible.

Weaknesses:

1. The experimental design is insufficient. The authors should include more baselines, especially some Riemannian optimization methods, like [1][2].

2. The theoretical analysis relies on gradient descent, while the experiments use AdamW. The authors should analyze the gap between these two optimizers.

3. The authors should add an analysis on rank selection.

[1] Bogachev, V., Aletov, V., Molozhavenko, A., Bobkov, D., Soboleva, V., Alanov, A., and Rakhuba, M. Riemannlora: A unified Riemannian framework for ambiguity-free lora optimization. arXiv preprint arXiv:2507.12142, 2025.
[2] Mo, Z., Huang, L.-K., and Pan, S. J. Parameter and memory efficient pretraining via low-rank riemannian optimization. In The Thirteenth International Conference on Learning Representations, 2025.

---

> ### Author Rebuttal · Authors · 2026-03-30
>
> We thank the reviewer for acknowledging that the paper is "well-organized and easy-to-follow", with a "rigorous mathematical analysis", and for saying that this "work is solid". Thanks a lot also for the helpful suggestions on strengthening the empirical positioning.
>
> > **The experimental design is insufficient. The authors should include more baselines, especially some Riemannian optimization methods, like [1][2].**
>
>
> > **The authors should add an analysis on rank selection.**
>
> Thanks a lot for the comments and relevant references. We **have performed two rank sensitivity analyses and added several baselines to our experiments**, including the Riemannian method LORO (Parameter and Memory Efficient Pretraining via Low-rank Riemannian Optimization, Mo et al, ICLR 2025). Here is the detail of the additional experiments:
> - Dataset sweep on Qwen2.5-3B. We evaluate LoRA, BaLoRA, DoRA, OLoRA, LoRA-RITE, RefLoRA, and LORO across Alpaca, Code-Feedback, OpenHermes, OpenOrca, and WizardLM. For each method, the learning rate and initialization scaling are selected from a 10×10 grid sweep on MetaMathQA, then applied to all datasets. In that experiment, most methods achieve very similar eval losses; **Balora is consistently among the top performing methods**.
> - Rank sweep on Qwen2.5-3B. We vary the LoRA rank in {2, 4, 8, 16, 32, 64, 128} with per-method optimal hyperparameters fixed as above. We run this on two datasets, Arxiv and DM-Mathematics, training for 1B tokens. In that experiment, **Balora achieves significantly better results than the other methods for the larger ranks** (64, 128). This illustrates the potential of our method for fine-tuning with large rank and lots of data.
>
> The results are at https://anonymous.4open.science/r/ICML_2026_rebuttal-D412/icml2026_rebuttal.pdf
>
> > **the authors should analyze the gap between using GD and AdamW in BaLoRA**
>
> Theoretically analyzing the convergence of Adam is still out of reach in the modern literature, as Adam lacks convergence guarantees even for convex quadratic objectives (Reddi, Kale, Kumar 2018). However, in the vanishing-momentum limit $x_{t+1} = x_t -\tau\, \mathrm{sign}(\nabla f(x_t))$, which is the prominent theoretical model for Adam, a simple argument shows that having a small conditioning (which is achieved for balanced minimizers independently of the optimizer, according to our theoretical results) allows to reach a smaller loss value. Indeed, if $x^\star$ is a nondegenerate minimizer and $H = D^2f(x^\star)$, writing $f(x)\approx \frac12 (x - x^\star)^\top H (x-x^\star) + f(x^\star)$ in the asymptotic regime gives, for $s_t:= \mathrm{sign}(H(x_t-x^\star))$:
> $$f(x_{t+1}) - f(x_t) \approx \tau \lVert H(x_t - x^\star)\rVert_1 +\frac{\tau^2}{2}s_t^\top H s_t \le -\tau\sqrt{2\lambda_\infty^{\mathrm{min}}(H) f(x_t)} + \frac{\tau^2}{2}\lambda_\infty^\mathrm{max}(H),$$
> where $\lambda_\infty^{\mathrm{min}}(H):=\min_{x\neq 0} \frac{\lVert Hx\rVert_1}{\lVert x\rVert_\infty}$ (replace min by max for $\lambda_\infty^\mathrm{max}$). Accordingly, the relevant notion is not asymptotic contraction to the minimizer, as in ordinary GD, but the terminal oscillation floor: the error level at which the negative drift and the positive discretization term become comparable, so that the iterates stop decreasing monotonically and begin to oscillate around $x^\star$. Monotone decrease is ensured as long as $f(x_t)>\tau^2 d\lambda_\infty^\mathrm{max}(H)\kappa(H)$  in dimension $d$, where $\kappa(H)$ is the spectral conditioning of $H$. Hence, a smaller $\kappa(H)$ is associated with a smaller minimum error floor reachable by signed-gradient descent.
>
> Many thanks for this question: we will add this discussion in the revised version to **significantly broaden the impact of our theoretical results**.
>
> We thank you again for your questions and comments, which help us improve the paper.

---

> > ### Author Rebuttal · Reviewer_dK5e · 2026-04-03
> >
> > Thanks the authors for additional experimental results and explanations for my concerns. I will keep my positive scores.

---

> > > ### Author Response · Authors · 2026-04-03
> > >
> > > Dear reviewer,
> > >
> > > We are happy to read this. Thanks a lot for your insightful remarks.

---

### Decision · Program_Chairs · 2026-04-30

**Decision:**

Accept (regular)

**Comment:**

This paper propose BaLoRA, which will project LoRA onto a balanced manifold to improve conditioning. Reviewers agreed that the paper is technically sound, clearly written, and offers a meaningful theoretical contribution connecting balanced factorizations to improved conditioning and faster convergence.

The main concerns are regarding the breadth of validation, related work, and the theoretical gap when Adam is used in practice. In the rebuttal, the authors added further comparisons, clarified the relation to some related work, and addressed most reviewer concerns. One reviewer still noted that a guidance for practitioner regarding different balanced variants is more beneficial, and the lack of a formal Adam analysis remains a limitation. Despite the remaining concerns, overall, I think the merit overweight the concerns, and recommend **acceptance**.